# Near-Universal Multiplicative Updates for Nonnegative Einsum Factorization

**John Hood** [1]   **Aaron Schein** [1 2]

## Abstract

Despite the ubiquity of multiway data across scientific domains, there are few performant and user-friendly methods that fit non-standard nonnegative tensor factorization models tailored to the data at-hand. Researchers may use gradient-based automatic differentiation, which often struggles under nonnegative constraints, choose between a limited set of methods with mature implementations, or implement their own model from scratch. As an alternative, we introduce NNEinFact, an einsum-based multiplicative update algorithm that fits any nonnegative tensor factorization expressible as a tensor contraction by minimizing one of many user-specified loss functions, including the $(\alpha, \beta)$-divergence. To use NNEinFact, the researcher specifies their model with a string. NNEinFact converges to a stationary point of the loss, supports missing data, and fits to tensors with hundreds of millions of entries in seconds. Empirically, NNEinFact fits custom models which outperform standard ones in prediction tasks on real-world tensor data by over $37\%$ and attains less than half the test loss of gradient-based methods while converging up to 90 times faster. Software is publicly available at github.com/jhood3/einfact.

## 1. Introduction

Matrix and tensor factorization models serve as fundamental tools for extracting latent structure from high-dimensional multi-way data (Kolda & Bader, 2009; Cichocki et al., 2015). These techniques impose structural constraints—such as low-rank factorizations or sparsity—to compress complex datasets while preserving their essential characteristics. Nonnegative variants of these factorizations (Cichocki et al., 2009; Chi & Kolda, 2012) have proven particularly valuable in scientific applications due to their interpretable, parts-based representations, leading to routine use for exploratory and descriptive data analysis.

A researcher's choice of factorization model crucially impacts interpretability, ability to recover underlying latent structure, and fit to the data (Kim & Choi, 2007; Kolda & Bader, 2009). Yet scientists face a critical gap: tailored factorizations—essential for capturing domain-specific structure—remain inaccessible to most researchers lacking considerable technical skills. Beyond a handful of algorithms with mature implementations, inference algorithms are typically tailored to individual models, implemented from scratch, scale poorly to large datasets, or require substantial implementation effort and programming ability. General methods based on automatic differentiation typically don't work well in practice, suffering from slow convergence, sensitivity to hyperparameter selection, among other problems (Shalev-Shwartz et al., 2017), making it difficult and time-consuming to explore novel models.

This paper develops NNEinFact, a method designed to bridge this gap. Centering the einsum operation, the computational backbone of numerous modern machine learning models (Paszke et al., 2019; Harris et al., 2020; Peharz et al., 2020), NNEinFact fits a wide family of nonnegative tensor factorizations under a general set of loss functions and is remarkably easy to use; see Figure 1.

**Contributions.** The rest of this paper introduces NNEinFact as a general tool for tailored nonnegative tensor factorization. We make the following contributions:

i.   **NNEinFact.** A simple, general-purpose nonnegative tensor factorization method built around three calls to the einsum operation that fits a wide family of models under a general set of loss functions.

ii.  **Flexible modeling.** Switching between loss functions involves changing one of two parameters, and specifying a tailored tensor decomposition model is done with a string — allowing the user to efficiently build, fit and refine models (Box, 1976; Blei, 2014).

iii. **Theoretical guarantees.** We use a majorization-minimization framework to prove NNEinFact's convergence to a stationary point of the objective.

iv.  **Empirical performance.** NNEinFact quickly esti-

---

[1]Department of Statistics, University of Chicago, IL, USA. [2]Data Science Institute, University of Chicago, IL, USA. Correspondence to: John Hood <johnhood@uchicago.edu>.

*Proceedings of the $43^{rd}$ International Conference on Machine Learning*, Seoul, South Korea. PMLR 306, 2026. Copyright 2026 by the author(s).

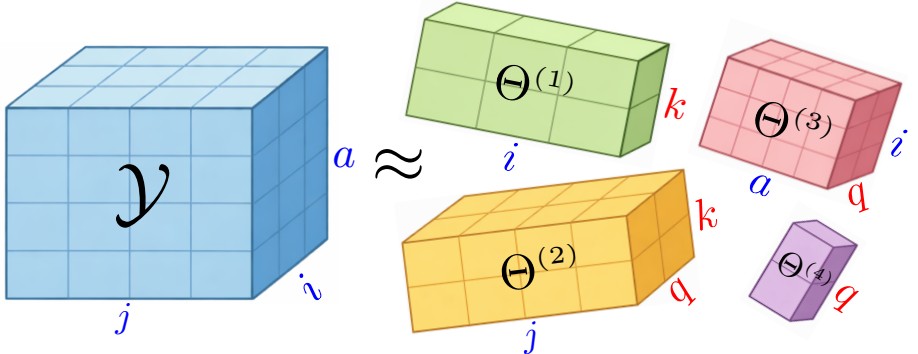

```python
from einfact import NNEinFact

model = NNEinFact(
    model_str='ik,jqk,aiq,q->ija',
    loss='alpha-beta', α=0.1, β=1.1)

model.fit(data=Y_ija)

θ_ik, θ_jqk, θ_aiq, θ_q = model.get_params()
```

**Model examples**

| | |
|---|---|
| NMF: | ik,jk→ij |
| Tri-NMF: | ic,cd,jd→ij |
| CP (3-mode): | ik,jk,ak→ija |
| CP (4-mode): | ik,jk,ak,tk→ijat |
| Tucker (3-mode): | ic,jd,ak,cdk→ija |
| Tucker (4-mode): | ic,jd,ak,cdkr→ijat |
| Low rank Tucker: | ic,jd,ak,cq,dq,kq→ija |
| Tensor train: | ic,jcd,ad→ija |
| Custom: | ic,jc,ad,dc→ija |
| | ic,jd,ack,kd,k→ija |

**Loss examples**

Euclidean distance
Forward KL divergence
Reverse KL divergence
Itakura-Saito divergence
Hellinger distance
Neyman $\chi^2$ divergence
Pearson $\chi^2$ divergence
$\alpha$-divergence
$\beta$-divergence
$(\alpha,\beta)$-divergence

*Figure 1.* **Schematic diagram highlighting NNEinFact's simplicity and generality.** Top left: running NNEinFact requires very few lines of code. Bottom left: an example custom tensor decomposition model. The top right panel provides a non-exhaustive list of examples of models that NNEinFact can fit; the bottom right panel provides a non-exhaustive list of loss functions that NNEinFact accommodates.

mates custom tensor decompositions, outperforming gradient-based automatic differentiation (the only practical alternative for estimating such models). These custom models attain substantially lower heldout loss than more common modeling choices.

v. **Interpretable representations.** In a rideshare pickup case study, we show how NNEinFact efficiently extracts interpretable spatiotemporal structure.

Together, these contributions offer a fresh perspective on modern nonnegative tensor factorization, serving as a foundation for future research in scalable, interpretable, and statistically principled scientific modeling and discovery.

## 2. Reformulating Generalized Tensor Factorization as Einsum Factorization

We consider the $M$-mode tensor $\mathcal{Y} \in \mathbb{R}_{\geq 0}^{I_1 \times \cdots \times I_M}$ (where mode $m$ has dimension $I_m$) with nonnegative entries $y_{i_1,\ldots,i_M}$, writing $\mathbf{i} = (i_1, \ldots, i_M)$. The task of nonnegative tensor decomposition is to approximate $\mathcal{Y}$ with $\widehat{\mathcal{Y}}$ such that $y_{\mathbf{i}} \approx \hat{y}_{\mathbf{i}}$ for all $\mathbf{i}$. We construct $\widehat{\mathcal{Y}}$ as the *tensor contraction* over $K$ contracted modes, where the $k^{\text{th}}$ contracted mode has dimension $R_k$. Writing $\mathbf{r} = (r_1, \ldots, r_K)$ to denote these additional indices, this is simply $\hat{y}_{\mathbf{i}} = \sum_{\mathbf{r}} \hat{y}_{\mathbf{i},\mathbf{r}}$.

We parameterize $\widehat{\mathcal{Y}}$ using $L$ vectors, matrices, or higher

order tensors $\Theta^{(1)}, \Theta^{(2)}, \ldots, \Theta^{(L)}$. For each $\Theta^{(\ell)}$, its modes may be partitioned into observed modes which are shared with $\mathcal{Y}$ and contracted modes which are not shared by $\mathcal{Y}$. We use $\mathbf{i}_\ell$ to index the observed indices of $\Theta^{(\ell)}$ and $\mathbf{r}_\ell$ to index its contracted indices. The family of decompositions we consider, referred to as *generalized tensor factorizations* (Yılmaz et al., 2011), takes the element-wise form

$$\hat{y}_{\mathbf{i}} = \sum_{\mathbf{r}} \hat{y}_{\mathbf{i},\mathbf{r}}, \quad \hat{y}_{\mathbf{i},\mathbf{r}} = \prod_{\ell=1}^{L} \theta^{(\ell)}_{\mathbf{i}_\ell,\mathbf{r}_\ell}. \quad (1)$$

Expression (1) is compactly written using *einsum notation*. Under einsum notation, indices which appear on the only left (and not the right) are summed over; the indices on the right specify the observed indices. Conveniently,

$$\mathbf{i}_1\mathbf{r}_1, \mathbf{i}_2\mathbf{r}_2, \ldots, \mathbf{i}_L\mathbf{r}_L \to i_1 i_2 \ldots i_M \quad (2)$$

corresponds to the einsum notation for (1) and is compactly expressed as a string in Python. We denote (2) by `model_str`. Given parameters $\{\Theta^{(\ell)}\}_{\ell=1}^{L}$, the operation

$$\widehat{\mathcal{Y}} \leftarrow \texttt{einsum}(\texttt{model\_str}, \{\Theta^{(\ell)}\}_{\ell=1}^{L}) \quad (3)$$

efficiently computes $\widehat{\mathcal{Y}}$. The family of generalized tensor or *einsum* factorizations includes the canonical choices of Tucker (Tucker, 1966), CP (Hitchcock, 1927), and tensor-train (Oseledets, 2011), among others.

**Example 1: Tucker.** The Tucker decomposition of multi-rank $(R_1, R_2, \ldots, R_M)$ is defined as

$$\hat{y}_{\mathbf{i}} = \sum_{r_1=1}^{R_1} \cdots \sum_{r_M=1}^{R_M} \hat{y}_{\mathbf{i},\mathbf{r}}, \quad \hat{y}_{\mathbf{i},\mathbf{r}} = \theta_{\mathbf{r}} \prod_{m=1}^{M} \theta_{i_m,r_m}^{(m)}. \quad (4)$$

It consists of factor matrices $\Theta^{(m)} \in \mathbb{R}^{I_m \times R_m}$ and *core tensor* $\Theta \in \mathbb{R}^{R_1 \times \cdots \times R_M}$. The model string is given by

$$i_1 r_1, \ldots, i_M r_M, r_1 \ldots r_M \rightarrow i_1 \ldots i_M.$$

**Example 2: CP.** The rank-$R$ CP decomposition is a special case of Tucker, where each mode has latent dimension $R_m = R$, and the core tensor has elements $\theta_{r_1,\ldots,r_M} = 1$ along the diagonal and 0 otherwise. It takes the form

$$\hat{y}_{\mathbf{i}} = \sum_{r=1}^{R} \hat{y}_{\mathbf{i},r}, \quad \hat{y}_{\mathbf{i},r} = \prod_{m=1}^{M} \theta_{i_m,r}^{(m)} \quad (5)$$

and has model string given by

$$i_1 r, \ldots, i_M r \rightarrow i_1 \ldots i_M.$$

**Example 3: Tensor-train.** The tensor-train decomposition, with $R_1 = R_{M+1} = 1$, is given by the string

$$i_1 r_1 r_2, i_2 r_2 r_3 \ldots, i_M r_M r_{M+1} \rightarrow i_1 \ldots i_M$$

and takes the form

$$\hat{y}_{\mathbf{i}} = \sum_{r_1=1}^{R_1} \cdots \sum_{r_{M+1}=1}^{R_{M+1}} \hat{y}_{\mathbf{i},\mathbf{r}}, \quad \hat{y}_{\mathbf{i},\mathbf{r}} = \prod_{m=1}^{M} \theta_{i_m,r_m,r_{m+1}}^{(m)}. \quad (6)$$

While these examples correspond to common decompositions, we emphasize how this family extends *beyond* them.

**Custom examples.** Recent work (Aguiar et al., 2024; Hood & Schein, 2024) develops variants of the nonnegative Tucker decomposition to model complex network data. Tucker has conceptual appeal, yet it suffers from the curse of dimensionality in its core tensor, which scales exponentially in its elements with the number of modes $M$. Further parameterizing the core tensor by a rank-$R$ CP decomposition such that

$$\theta_{\mathbf{r}} = \sum_{r=1}^{R} \prod_{m=1}^{M} \theta_{r_m r}^{(M+m)} \quad (7)$$

reduces the number of core tensor parameters from $\prod_{m=1}^{M} R_m$ to $R(\sum_{m=1}^{M} R_m)$, a quantity linear in $M$. This modification corresponds to the string

$$i_1 r_1, \ldots, i_M r_M, r_1 r, \ldots, r_M r \rightarrow i_1 \ldots i_M.$$

We can construct other decompositions, such as the many-body approximation (Ghalamkari et al., 2023), which consists of matrices corresponding to pairs of observed indices. For the three-mode setting,

$$i_1 i_2, i_2 i_3, i_1 i_3 \rightarrow i_1 i_2 i_3.$$

*Tensor networks*, such as the tensor ring (Zhao et al., 2016) and tensor wheel (Wu et al., 2022) decompositions may also be expressed in this form. In the three-mode setting, tensor ring (TR) and tensor wheel (TW) take the form

$$i_1 r_1 r_2, i_2 r_2 r_3, i_3 r_3 r_1 \rightarrow i_1 i_2 i_3 \quad \text{(TR)}$$
$$r_1 r_2 r_3, i_1 r_1 r_4 r_5, i_2 r_2 r_5 r_6, i_3 r_3 r_4 r_6 \rightarrow i_1 i_2 i_3 \quad \text{(TW)}$$

Additionally, the custom models

$$i_1 r_1, i_2 i_3 r_1 \rightarrow i_1 i_2 i_3$$
$$i_1, i_2 r_1, i_3 r_1 \rightarrow i_1 i_2 i_3$$
$$i_1 r_1, i_2 r_2, i_3 r_2, r_1 r_2 \rightarrow i_1 i_2 i_3$$

are all members of this family. Given its scope, we study the problem of estimating *any* factorization of the form (1).

## 3. Near-Universal Multiplicative Updates

For any parameterization of $\widehat{\mathcal{Y}}$ by expression (1) and loss function $\mathcal{L}$, we consider the minimization problem

$$\min_{\Theta^{(1)}, \ldots, \Theta^{(L)}} \mathcal{L}(\mathcal{Y}, \widehat{\mathcal{Y}}) = \sum_{\mathbf{i}} \mathcal{L}(y_{\mathbf{i}}, \hat{y}_{\mathbf{i}}), \quad \Theta^{(\ell)} \geq \epsilon \quad (8)$$

(for a very small $\epsilon > 0$) to estimate $\{\Theta^{(\ell)}\}_{\ell=1}^{L}$. The general nonconvexity of (8) motivates an iterative algorithm that updates $\Theta^{(\ell)}$ while fixing $\Theta^{(\ell')}$ for all $\ell' \neq \ell$. Objective (8) is broad; we consider differentiable loss functions which satisfy a certain decomposability property stated in Theorem 3.1. This decomposability property ensures the convergence of Algorithm 1.

**Theorem 3.1.** *Suppose that $\mathcal{L}(x,y)$ is differentiable in its second argument with partial derivative map $\partial_y \mathcal{L}$, has a convex-concave decomposition*

$$\mathcal{L}(x,y) = \mathcal{L}^{vex}(x,y) + \mathcal{L}^{cave}(x,y) \quad (9)$$

*with respect to $y$, and satisfies the decomposability property*

$$\frac{\partial_y \mathcal{L}^{vex}(x, \lambda y) + \partial_y \mathcal{L}^{cave}(x,y)}{c(\lambda)} = g(\lambda) b(x,y) - a(x,y) \quad (10)$$

*for $\lambda > 0$, $a(x,y)$, $b(x,y)$, and $g(\lambda)$, where $g : \mathbb{R}^+ \rightarrow \mathbb{R}$ is invertible onto its image. Then $\mathcal{L}(\mathcal{Y}, \widehat{\mathcal{Y}})$ is non-increasing under the multiplicative update*

$$\Theta^{(\ell)} \leftarrow \max\left(\epsilon, \Theta^{(\ell)} \odot g^{-1}\left(\frac{\sum_{\mathbf{i}} [\nabla_{\Theta^{(\ell)}} \hat{y}_{\mathbf{i}}] a(y_{\mathbf{i}}, \hat{y}_{\mathbf{i}})}{\sum_{\mathbf{i}} [\nabla_{\Theta^{(\ell)}} \hat{y}_{\mathbf{i}}] b(y_{\mathbf{i}}, \hat{y}_{\mathbf{i}})}\right)\right), \quad (11)$$

*provided that the argument of $g^{-1}$ in (11) lies in $\mathrm{Range}(g)$.*

A quick evaluation of (10) shows that many loss functions possess the decomposability property, including the $(\alpha, \beta)$-divergence (Cichocki & Amari, 2010), which includes

the squared Euclidean distance, KL divergence, reverse KL divergence, Itakura–Saito divergence, $\alpha$-divergence, $\beta$-divergence (Basu et al., 1998), squared Hellinger distance, Pearson and Neyman $\chi^2$ divergences as special cases. As such, our work unifies and extends much existing work; we elaborate on these connections in Section 5. Beyond these, the Bernoulli, binomial, negative binomial, and geometric distributions all have negative log-likelihoods which satisfy this form of decomposability under specific reparameterizations outlined in Appendix B.

We defer a proof of monotonicity to Section 4 and instead focus on computing the expression in (11) given by

$$g^{-1}\left(\tfrac{\mathtt{A}}{\mathtt{B}}\right) := g^{-1}\left(\frac{\sum_{\mathbf{i}}[\nabla_{\Theta^{(\ell)}}\hat{y}_{\mathbf{i}}]a(y_{\mathbf{i}},\hat{y}_{\mathbf{i}})}{\sum_{\mathbf{i}}[\nabla_{\Theta^{(\ell)}}\hat{y}_{\mathbf{i}}]b(y_{\mathbf{i}},\hat{y}_{\mathbf{i}})}\right).$$

Given the tensors $\mathcal{Y}$ and $\widehat{\mathcal{Y}}$, the loss-dependent functions $b(y_{\mathbf{i}},\hat{y}_{\mathbf{i}})$, $a(y_{\mathbf{i}},\hat{y}_{\mathbf{i}})$ and $g^{-1}(x)$ are often remarkably simple and cheap to compute. For example, the least-squares loss yields $b(y_{\mathbf{i}},\hat{y}_{\mathbf{i}}) = \hat{y}_{\mathbf{i}}$, $a(y_{\mathbf{i}},\hat{y}_{\mathbf{i}}) = y_{\mathbf{i}}$, and $g(x) = x$.

We can compute $\widehat{\mathcal{Y}}$ using the einsum expression given in (3). Crucially, the numerator and denominator of (11) are *also* neatly expressible using einsum. Consider the numerator $\mathtt{A} = \sum_{\mathbf{i}}[\nabla_{\Theta^{(\ell)}}\hat{y}_{\mathbf{i}}]a(y_{\mathbf{i}},\hat{y}_{\mathbf{i}})$. $\nabla_{\Theta}\hat{y}_{\mathbf{i}}$ has element-wise form

$$\frac{\partial \hat{y}_{\mathbf{i}}}{\partial \theta_{\mathbf{i}_\ell,\mathbf{r}_\ell}} = \sum_{\mathbf{r}}\mathbf{1}(\mathbf{i}_\ell \subseteq \mathbf{i}, \mathbf{r}_\ell \subseteq \mathbf{r})\prod_{\ell' \neq \ell}\theta_{\mathbf{i}_{\ell'},\mathbf{r}_{\ell'}}^{(\ell')}. \qquad (12)$$

Plugging this into the numerator yields the expression

$$\sum_{\mathbf{i},\mathbf{r}}a(y_{\mathbf{i}},\hat{y}_{\mathbf{i}})\mathbf{1}(\mathbf{i}_\ell \subseteq \mathbf{i}, \mathbf{r}_\ell \subseteq \mathbf{r})\prod_{\ell' \neq \ell}\theta_{\mathbf{i}_{\ell'},\mathbf{r}_{\ell'}}^{(\ell')}. \qquad (13)$$

This is an einsum. Defining the string $\mathtt{einstr}_\ell$ as

$$\mathtt{einstr}_\ell := \mathbf{i}_1\mathbf{r}_1,\dots,\mathbf{i}_{\ell-1}\mathbf{r}_{\ell-1},\mathbf{i},$$
$$\mathbf{i}_{\ell+1}\mathbf{r}_{\ell+1},\dots,\mathbf{i}_L\mathbf{r}_L \to \mathbf{i}_\ell\mathbf{r}_\ell,$$

expression (13) is exactly expressed as

$$\mathtt{A} = \mathtt{einsum}(\mathtt{einstr}_\ell,$$
$$\Theta^{(1)},\dots,\Theta^{(\ell-1)},a(\mathcal{Y},\widehat{\mathcal{Y}}),\Theta^{(\ell+1)},\dots,\Theta^{(L)}).$$

The denominator's corresponding einsum is nearly identical:

$$\mathtt{B} = \mathtt{einsum}(\mathtt{einstr}_\ell,$$
$$\Theta^{(1)},\dots,\Theta^{(\ell-1)},b(\mathcal{Y},\widehat{\mathcal{Y}}),\Theta^{(\ell+1)},\dots,\Theta^{(L)}).$$

Here, $a(\cdot)$ and $b(\cdot)$ are applied element-wise. To create $\mathtt{einstr}_\ell$, we also implement $\mathtt{swap}(\mathtt{model\_str},\ell)$ which swaps the model output $\mathbf{i}$ with the $\ell^{\text{th}}$ entry $\mathbf{i}_\ell\mathbf{r}_\ell$. The full iterative algorithm is given in Algorithm 1, which applies update (11) to each $\Theta^{(\ell)}$ until convergence.

---

**Algorithm 1** NNEinFact: multiplicative update algorithm

**input** observed tensor $\mathcal{Y}$, model string $\mathtt{model\_str}$, initial parameters $\{\Theta^{(1)},\dots,\Theta^{(L)}\}$, loss function $\mathcal{L}$, $\epsilon$
1: **for** $\ell = 1,\dots,L$ **do**
2:    $\mathtt{einstr}_\ell \leftarrow \mathtt{swap}(\mathtt{model\_str},\ell)$
3: **end for**
4: **while** not converged **do**
5:    **for** $\ell = 1,\dots,L$ **do**
6:      $\widehat{\mathcal{Y}} \leftarrow \mathtt{einsum}(\mathtt{model\_str},\{\Theta^{(\ell)}\}_{\ell=1}^L)$
7:      $\mathtt{A} \leftarrow \mathtt{einsum}(\mathtt{einstr}_\ell,$
       $\Theta^{(1)},\dots,\Theta^{(\ell-1)},a(\mathcal{Y},\widehat{\mathcal{Y}}),\Theta^{(\ell+1)},\dots,\Theta^{(L)})$
8:      $\mathtt{B} \leftarrow \mathtt{einsum}(\mathtt{einstr}_\ell,$
       $\Theta^{(1)},\dots,\Theta^{(\ell-1)},b(\mathcal{Y},\widehat{\mathcal{Y}}),\Theta^{(\ell+1)},\dots,\Theta^{(L)})$
9:      $\Theta^{(\ell)} \leftarrow \min\left(\epsilon, \Theta^{(\ell)} \odot g^{-1}\left(\tfrac{\mathtt{A}}{\mathtt{B}}\right)\right)$
10:    **end for**
11: **end while**
   **return** $\{\Theta^{(1)},\dots,\Theta^{(L)}\}$

---

**Missing data.** Training and evaluation with missing values is straightforward. Suppose that $\mathcal{Y}$ consists of missing observations, and denote the set of observed (not missing) indices as $\mathcal{O}$. The objective in this setting depends on the set of observed indices only:

$$\min_{\Theta^{(1)},\dots,\Theta^{(L)}}\mathcal{L}(\mathcal{Y},\widehat{\mathcal{Y}}) = \sum_{\mathbf{i} \in \mathcal{O}}\mathcal{L}(y_{\mathbf{i}},\hat{y}_{\mathbf{i}}), \quad \Theta^{(\ell)} \geq \epsilon.$$

As such, defining the binary mask $\mathcal{M}$ the size of $\mathcal{Y}$, where

$$\mathcal{M}_{\mathbf{i}} = \begin{cases} 1 & \text{if } \mathbf{i} \in \mathcal{O} \\ 0 & \text{otherwise,} \end{cases} \qquad (14)$$

one proceeds as usual by replacing $\mathcal{Y}$ and $\widehat{\mathcal{Y}}$ with $\mathcal{M} \odot \mathcal{Y}$ and $\mathcal{M} \odot \widehat{\mathcal{Y}}$ in Algorithm 1. Here, $\odot$ denotes the Hadamard product, or element-wise multiplication.

## 4. Theoretical Guarantees

We use a majorization-minimization (MM) framework to establish Algorithm 1's convergence to a stationary point of the loss $\mathcal{L}$. Given a function to minimize ($\mathcal{L}$), MM offers a principled approach to optimization by iteratively constructing tight upper bounds, or surrogate functions, $Q$, that *majorize* $\mathcal{L}$ and minimizing them. Iteratively majorizing $\mathcal{L}$ and minimizing $Q$ yields a convergent algorithm that monotonically decreases $\mathcal{L}$.

Formally, the function

$$Q : \text{dom}(\Theta) \times \text{dom}(\Theta) \to \mathbb{R}_{\geq 0}$$

is a *surrogate function* to $\mathcal{L}$ iff for all $\Theta^{(\ell)}, \widetilde{\Theta}^{(\ell)} \in \text{dom}(\Theta)$,

$$Q(\Theta^{(\ell)} \mid \widetilde{\Theta}^{(\ell)}) \geq \mathcal{L}(\Theta^{(\ell)}) = Q(\Theta^{(\ell)} \mid \Theta^{(\ell)}). \qquad (15)$$

The surrogate property ensures that at each step, minimizing $Q$ decreases $\mathcal{L}$. By decomposing $\mathcal{L}$ into convex and concave components as in equation (9) and upper bounding each component individually, we construct a surrogate function to $\mathcal{L}$. The exact form of $Q$ is given in Lemma 4.1.

**Lemma 4.1.** *Consider the differentiable convex-concave decomposition of $\mathcal{L}(x, y)$ in equation (9) and define $\tilde{y}_{\mathbf{i}, \mathbf{r}_\ell}$ as the sum over latent indices $r_k \notin \mathbf{r}_\ell$:*

$$\tilde{y}_{\mathbf{i}, \mathbf{r}_\ell} = \sum_{\mathbf{r} \setminus \mathbf{r}_\ell} \tilde{y}_{\mathbf{i}, \mathbf{r}}, \quad such\ that \quad \sum_{\mathbf{r}_\ell} \tilde{y}_{\mathbf{i}, \mathbf{r}_\ell} = \sum_{\mathbf{r}} \tilde{y}_{\mathbf{i}, \mathbf{r}} = \tilde{y}_{\mathbf{i}}.$$

*Then the function*

$$Q(\Theta^{(\ell)} \mid \widetilde{\Theta}^{(\ell)}) = \sum_{\mathbf{i}, \mathbf{r}_\ell} \frac{\tilde{y}_{\mathbf{i}, \mathbf{r}_\ell}}{\tilde{y}_{\mathbf{i}}} \mathcal{L}^{vex}\left(y_{\mathbf{i}}, \tilde{y}_{\mathbf{i}} \frac{\theta^{(\ell)}_{\mathbf{i}_\ell, \mathbf{r}_\ell}}{\tilde{\theta}^{(\ell)}_{\mathbf{i}_\ell, \mathbf{r}_\ell}}\right) \quad (16)$$
$$+ \sum_{\mathbf{i}} \mathcal{L}^{cave}(y_{\mathbf{i}}, \tilde{y}_{\mathbf{i}}) + \partial_y \mathcal{L}^{cave}(y_{\mathbf{i}}, \tilde{y}_{\mathbf{i}})(\hat{y}_{\mathbf{i}} - \tilde{y}_{\mathbf{i}})$$

*is a surrogate function to $\mathcal{L}(\Theta^{(\ell)})$.*

We prove this result in Appendix A. Moreover, $Q$ is easily minimized: Lemma 4.2 establishes that it attains a minimum under the multiplicative update in (11) applied to $\tilde{\Theta}^{(\ell)}$.

**Lemma 4.2.** $Q(\Theta^{(\ell)} \mid \widetilde{\Theta}^{(\ell)})$ *is convex in $\Theta^{(\ell)}$. If property (10) holds, then $Q(\Theta^{(\ell)} \mid \widetilde{\Theta}^{(\ell)})$ is minimized by*

$$\Theta^{(\ell)} = \max\left(\epsilon, \widetilde{\Theta}^{(\ell)} \odot g^{-1}\left(\frac{\sum_{\mathbf{i}}[\nabla_{\Theta^{(\ell)}} \hat{y}_{\mathbf{i}}] a(y_{\mathbf{i}}, \tilde{y}_{\mathbf{i}})}{\sum_{\mathbf{i}}[\nabla_{\Theta^{(\ell)}} \hat{y}_{\mathbf{i}}] b(y_{\mathbf{i}}, \tilde{y}_{\mathbf{i}})}\right)\right)$$

*where $g^{-1}$ and $/$ are applied element-wise.*

See Appendix A for the proof. The minimizer matches the einsum-based multiplicative update of (11). Thus, we may establish Algorithm 1's convergence.

**Theorem 4.3.** *If prop. (10) holds, then the iterative process in Algorithm 1 converges. Every limit point of Algorithm 1 is a stationary point of objective 8.*

Algorithm 1 iterates over $\ell \in [L]$, setting

$$\Theta^{(\ell)} \leftarrow \arg\min_\Theta Q(\Theta, \widetilde{\Theta}^{(\ell)}). \quad (17)$$

At each iteration,

$$\mathcal{L}(\widetilde{\Theta}^{(\ell)}) = Q(\widetilde{\Theta}^{(\ell)} \mid \widetilde{\Theta}^{(\ell)}) \underset{(1)}{\geq} Q(\Theta^{(\ell)} \mid \widetilde{\Theta}^{(\ell)})$$
$$\underset{(2)}{\geq} Q(\Theta^{(\ell)} \mid \Theta^{(\ell)}) = \mathcal{L}(\Theta^{(\ell)}) \geq 0.$$

Inequality (1) follows from the minimization of $Q$ and inequality (2) follows from the surrogate property (15). Monotonicity and boundedness imply convergence of the objective values. In Appendix A we show that the limit points are stationary points of objective (8).

MM provides a powerful framework for establishing monotonicity and convergence; we are not the first to use it. The expectation-maximization (Dempster et al., 1977), iteratively reweighted least-squares (Holland & Welsch, 1977), and nonnegative matrix factorization algorithms (Lee & Seung, 2000) are all special instances of MM.

**Regularization.** The MM framework permits a straightforward extension of Algorithm 1 that incorporates regularization. Given a loss function $\mathcal{L}$ satisfying decomposability property (10), we consider penalized objectives of the form

$$\mathcal{L}(\mathcal{Y}, \widehat{\mathcal{Y}}) + \sum_{\ell=1}^{L} R^{(\ell)}(\Theta^{(\ell)}), \quad \Theta^{(\ell)} \geq \epsilon \quad (18)$$

for differentiable $R^{(\ell)}$. In Appendix A, we show that if we can write

$$\nabla_\Theta R(\Theta) = \nabla_\Theta R(\lambda \widetilde{\Theta}) = c(\lambda)[g(\lambda)\tilde{b}(\widetilde{\Theta}) - \tilde{a}(\widetilde{\Theta})],$$

then the corresponding multiplicative update is given by

$$\Theta^{(\ell)} = \min\left(\epsilon, \widetilde{\Theta}^{(\ell)} \odot g^{-1}\left(\frac{\tilde{a}(\widetilde{\Theta}^{(\ell)}) + \sum_{\mathbf{i}}[\nabla_{\Theta^{(\ell)}} \hat{y}_{\mathbf{i}}] a(y_{\mathbf{i}}, \hat{y}_{\mathbf{i}})}{\tilde{b}(\widetilde{\Theta}^{(\ell)}) + \sum_{\mathbf{i}}[\nabla_{\Theta^{(\ell)}} \hat{y}_{\mathbf{i}}] b(y_{\mathbf{i}}, \hat{y}_{\mathbf{i}})}\right)\right).$$

While this condition is not trivially satisfied, many common combinations of loss function and regularization penalty, such as (squared Euclidean distance loss, $\ell_2$ regularization) and (KL divergence loss, $\ell_1$ regularization), do satisfy it. We refer the reader to additional details in Appendix A, where we show how to naturally extend the decomposability property to accommodate many forms of regularized objectives.

## 5. Related Work

An extensive literature develops algorithms tailored to specific nonnegative tensor decompositions and loss functions. Mature implementations exist for CP and Tucker (Kim & Choi, 2007; Kim et al., 2008; Cichocki & Phan, 2009; Cichocki et al., 2009; Phan & Cichocki, 2011; Chi & Kolda, 2012; Zhou et al., 2015), with the MATLAB Tensor Toolbox (Bader & Kolda, 2006), Python's Tensorly library (Kossaifi et al., 2019), and R's nnTensor package (Tsuyuzaki & Nikaido, 2023) providing accessible open-source code. These libraries exclusively implement CP and Tucker. Alternative factorizations require either developing specialized algorithms for each model or relying on generic optimization methods that suffer from slow convergence and hyperparameter sensitivity (Bengio et al., 2017; Shalev-Shwartz et al., 2017), creating a significant barrier to exploring domain-specific tensor models.

Researchers have begun to exploit the general form of equation (1) to develop flexible tensor decomposition methods. Yılmaz & Cemgil (2010) introduce equation (1) as *probabilistic latent tensor factorization* and derive

multiplicative updates for the Euclidean distance. Yılmaz et al. (2011) extended this framework to *generalized tensor factorization* under the $\beta$-divergence family, deriving heuristic multiplicative updates based on the connection between exponential families and Bregman divergences. However, their updates lack formal convergence guarantees—they do not prove monotonic descent or convergence to a stationary point. Their approach is limited to $\beta$-divergence and lacks a scalable or accessible implementation. NNEinFact advances this work by: (i) providing a rigorous majorization-minimization-based proof of monotonicity and convergence, (ii) introducing the decomposability framework (10) that vastly expands the family of applicable loss functions, and (iii) centering the einsum function as a general and easily accessible tool amenable to GPU acceleration.

Recent work develops tensor methods for *discrete density estimation* under particular loss functions. Ghalamkari et al. (2026) provides an $\alpha$-divergence minimization framework for mixtures of CP, Tucker, and tensor-train. Ghalamkari et al. (2025) view normalized nonnegative tensors as discrete distributions and minimize KL divergence for decompositions under (1), deriving specific algorithms for CP, Tucker, and tensor-train. While more general, these approaches still develop specialized algorithms for particular model-loss combinations.

Beyond squared Euclidean distance and KL divergence, various f-divergences and Bregman divergences (Rényi, 1961; Bregman, 1967) have been shown to handle sparse and noisy tensor data well. The $\alpha$-divergence (an f-divergence) and $\beta$-divergence (a Bregman divergence) are robust to missing values, outliers, and model misspecification. Cichocki et al. (2007) and Févotte & Idier (2011) use the MM framework to develop multiplicative update algorithms for these divergences under NMF, CP and Tucker. Both divergences are special cases of the $(\alpha, \beta)$-divergence family, for which Cichocki et al. (2011) proposed multiplicative updates in the NMF setting. The decomposability property (10) unifies and extends these prior results to arbitrary einsum factorizations and novel loss functions.

## 6. Empirical Evaluation

We organize our experiments to compare model structures and optimization algorithms across a variety of loss functions. First, we compare NNEinFact to the natural baseline of gradient-based automatic differentiation, demonstrating superior model fit at a fraction of the cost. Second, we demonstrate that problem-tailored tensor decompositions achieve superior data fit compared to standard methods across a range of real-world settings. Last, through a case study on New York City Uber pickup data, we show how custom models recover interpretable, scientifically meaningful spatiotemporal structure using a very small parameter count.

### 6.1. Datasets

We consider three complex multi-way datasets from network science, a domain where practitioners are developing tensor decomposition methods to capture latent structure underlying the data (Contisciani et al., 2022; Aguiar et al., 2024; Hood et al., 2026). When handling real tensor data, modes typically correspond to a scientifically meaningful quantity (such as time). In such cases, we use the most natural letter to index that mode. For example, we index the 'time' mode by $t$ and the 'import' mode by $i$.

**ICEWS.** Dyadic relational data of the form "country $i$ took action $a$ toward country $j$ at time $t$" are commonly studied in international relations (Schrodt et al., 1995). These data can be interpreted as a directed, dynamic multilayer network comprising $V$ nodes (representing actors), $A$ layers (representing action types), and $T$ time periods. Such structure naturally corresponds to a 4-mode count tensor $\mathcal{Y} \in \mathbb{N}_0^{V \times V \times A \times T}$, where each element $y_{ijat}$ denotes the number of times country $i$ took action $a$ toward country $j$ during time period $t$. We analyze data of this form from the Integrated Crisis Early Warning System (ICEWS) (Boschee et al., 2015) spanning 1995–2013, where events are aggregated into monthly counts, yielding an observed tensor $\mathcal{Y}^{(\text{icews})} \in \mathbb{N}_0^{249 \times 249 \times 20 \times 228}$.

**Uber.** Using Uber ride pickup data (Smith et al., 2017), we construct a 5-mode spatiotemporal tensor $\mathcal{Y}^{(\text{uber})} \in \mathbb{N}_0^{27 \times 7 \times 24 \times 100 \times 100}$ where $y_{wdhij}$ denotes the number of ride pickups in hour $h$ of day $d$ of week $w$ at spatial location $(i, j)$. The dataset consists of ride pickups in New York City from April to September 2014. This dataset exhibits strong spatial dependencies and rich temporal structure, including seasonal and cyclical patterns.

**WITS.** Finally, we use merchandise-trade data accessed from the World Integrated Trade Solution (WITS) (World Integrated Trade Solution, 2025). WITS is a delivery platform maintained by the World Bank that provides standardized access to several primary sources. In this paper, we use export data as reported by national customs authorities via WITS as done by Jian et al. (2025). We retain annual trade values (thousand USD) for 96 HS2 categories across 196 countries over 1996–2024. We consider international trade data in the form of a 4-mode tensor $\mathcal{Y}^{(\text{trade})} \in \mathbb{N}_0^{196 \times 196 \times 96 \times 29}$, where $y_{eigt}$ represents the value (in U.S. dollars) of good $g$ exported from country $e$ to country $i$ in year $t$. This data suffers from missingness and noise, primarily due to asymmetric reporting from importing and exporting countries (Chen et al., 2022).

### 6.2. Experimental Details

We implement NNEinFact for the $(\alpha, \beta)$-divergence defined in Appendix B, denoting the loss by $\mathcal{L}_{\alpha,\beta}$. There

is a tight connection between $(\alpha, \beta)$-divergence minimization and maximum likelihood estimation in exponential family models (Yilmaz & Cemgil, 2012). In our experiments, we leverage this relationship to choose $\alpha$ and $\beta$, selecting $\beta = 0$ for count data (corresponding to the Poisson likelihood), $\beta = -0.5$ for sparse positive continuous data (corresponding to the compound Poisson-gamma likelihood). We further adjust $\alpha \in \{0.7, 0.8, 1.0, 1.2, 1.3\}$. When $\alpha, \beta, \alpha + \beta \neq 0$, as is the case in all of our experiments, the update-specific functions $a$ and $b$ take the form

$$a(x, y) = x^\alpha y^{\beta-1}$$
$$b(x, y) = y^{\alpha+\beta-1}$$

From this perspective, $\alpha$ is a robustness parameter: choices of $\alpha < 1$ reduce the signal of large values (such as outliers), while choices of $\alpha > 1$ amplify them, decreasing the signal of small values (such as missing values encoded as zeros).

**Baseline Models.** For all datasets, we fit a variety of common factorizations including CP and tensor-train as defined in equations (5) and (6). We also fit four versions of Tucker. We fit both *hypercubic* Tucker, where all latent dimensions $R_m$ are equal, and the Tucker decomposition where $R_m$ is proportional to the observed dimension $I_m$, as well as each of their low-rank (LR) variants, defined in equation (7). All models use a roughly equal number of parameters, which are initialized at random from the standard uniform distribution, and are fit using the multiplicative updates of Algorithm 1. The CP and Tucker models are also fit using two block-coordinate descent methods that minimize the squared Euclidean distance during training, HALS-CP (Cichocki & Phan, 2009) and HALS-Tucker (Phan & Cichocki, 2011).

**Custom models.** We also fit custom models tailored to each dataset. When designing these models, we use basic domain-specific knowledge to determine which modes have similar/different complexity and generally recommend applying domain-specific knowledge to design more complex models. For the Uber rideshare data, we fit the model

$$\hat{y}_{wdhij} = \sum_{r=1}^{R} \theta_{wr}^{(1)} \theta_{dr}^{(2)} \theta_{hr}^{(3)} \sum_{k=1}^{K} \theta_{irk}^{(4)} \theta_{jrk}^{(5)} \qquad (19)$$

corresponding to

$$\texttt{wr,dr,hr,irk,jrk} \rightarrow \texttt{wdhij}.$$

Each latent class $r$ corresponds to a different temporal pattern and each temporal pattern has an additional $K$ factors corresponding to the latitude and longitude modes $i$ and $j$. Setting $K = 1$ recovers the rank-$R$ CP decomposition, while $K > 1$ allocates relatively more parameters to the spatial modes than the others.

The ICEWS tensor has three modes with dimension greater than 200 (corresponding to sender, receiver, month), yet

the 'action' mode has dimension 20. Such oblong structure motivates our custom model. The rank $R$ CP decomposition assigns all modes the same latent dimension to create factor matrices with size $I_m \times R$. Instead, we consider the model:

$$\hat{y}_{ijat} = \sum_{r=1}^{R} \theta_{ir}^{(1)} \theta_{jr}^{(2)} \left( \sum_{k=1}^{K} \phi_{ak} w_{kr} \right) \theta_{tr}^{(4)}, \qquad (20)$$

a rank-$R$ CP decomposition with factor matrix $\Theta^{(3)} := \Phi W$ corresponding to the 'action' mode of rank $K \ll R$. This model corresponds to the string

$$\texttt{ir,jr,ak,kr,tr} \rightarrow \texttt{ijat}.$$

Finally, the WITS tensor includes 196 importing and exporting countries and 96 goods but only 29 time steps. We impose low-rank structure on the 'time' mode to estimate

$$\hat{y}_{eigt} = \sum_{r=1}^{R} \theta_{er}^{(1)} \theta_{ir}^{(2)} \theta_{gr}^{(3)} \sum_{k=1}^{K} \phi_{tk} w_{kr}. \qquad (21)$$

for $K \ll R$, which corresponds to the string

$$\texttt{er,ir,gr,tk,kr} \rightarrow \texttt{eigt}.$$

This model imposes less complex temporal structure than that of the network structure, which governs good-specific interactions between importers and exporters.

**Baseline algorithms.** We implement gradient-based automatic differentiation as a baseline to the multiplicative update algorithm in Section 3 and minimize the loss using Adam (Kingma & Ba, 2015). We update the log-transformed parameters to uphold the nonnegativity constraint. Adam's fit is often sensitive to its initial learning rate parameter, so we use baselines with initial learning rates $(0.01, 0.05, 0.1, 0.3, 0.5, 1.0)$. All algorithms were implemented in Pytorch and run on a GPU.

**Experimental design.** We split each dataset for training and evaluation. For each observed tensor, we create ten[1] train-test splits. For each split, we randomly assign each element **i** of $\mathcal{Y}$ to the training set with 90% probability and otherwise assign it to a heldout set $\mathcal{H}$. We further allocate 5% of the training set to a validation set $\mathcal{V}$ and use it to check for early stopping. Each method minimizes the training loss

$$\mathcal{L}^{\text{train}} := \sum_{\mathbf{i} \notin \mathcal{H}, \mathcal{V}} \mathcal{L}(y_\mathbf{i}, \hat{y}_\mathbf{i}).$$

**Evaluation metric.** We evaluate each method using average heldout loss, defined as

$$\mathcal{L}_{\alpha,\beta}^{\text{heldout}} := \frac{1}{|\mathcal{H}|} \sum_{\mathbf{i} \in \mathcal{H}} \mathcal{L}_{\alpha,\beta}(y_\mathbf{i}, \hat{y}_\mathbf{i}).$$

---

[1]For WITS, we create 50 train-test splits due to high variation in heldout loss among splits.

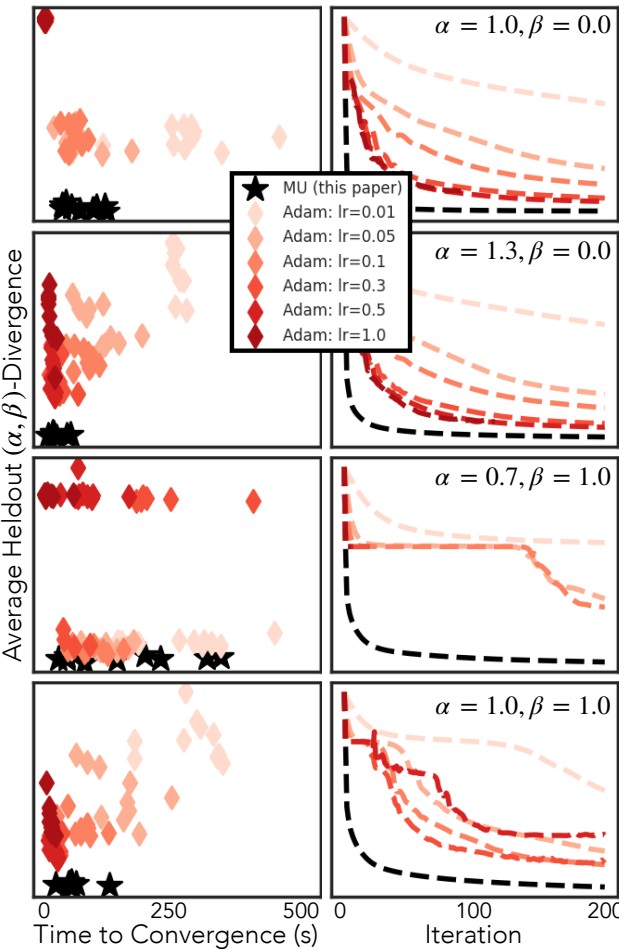

*Figure 2.* **NNEinFact is more efficient than automatic differentiation and achieves better fit.** NNEinFact's multiplicative update algorithm is shown in black, while the gradient-based automatic differentiation baselines are shown in red. Each row corresponds to a different $(\alpha, \beta)$ parameterization. On the left, each point corresponds to a random train-test split and method. Heldout $(\alpha, \beta)$-divergence is shown against wall-clock time to convergence. On the right column, we show heldout $(\alpha, \beta)$-divergence (on log scale) against training iteration. Each line represents one run of each algorithm, for different $(\alpha, \beta)$-divergences.

**Time to convergence.** We also compare each optimization method's runtime to convergence for many loss functions. In particular, we fit each dataset's custom model using the $(\alpha, \beta)$-divergence with $\alpha \in \{0.7, 1.0, 1.3\}$ and $\beta \in \{0, 1\}$, for a total of six distinct configurations. Each method runs until it meets the convergence criteria specified in Appendix B. We measure time to convergence using wall-clock time from initialization.

**Synthetic experiments.** In Appendix C, we consider two synthetic settings to evaluate NNEinFact's parameter recovery and tensor reconstruction abilities relative to a variety of baseline methods when the ground truth setting is known. We find that, when aligned with the ground truth,

NNEinFact exhibits superior performance in both settings and refer the reader to Appendix C for additional details.

### 6.3. Results

Figure 2 shows how NNEinFact's multiplicative updates outperform gradient-based automatic differentiation in both runtime and heldout loss across four of the six $(\alpha, \beta)$ parameterizations when fit to the Uber data. Each row corresponds to a different $(\alpha, \beta)$ parameterization. The left column shows the heldout loss against runtime to convergence (where lower is better); each plot's optimal region is the bottom left corner. Each point represents a different train-test split. We observe that NNEinFact's points typically congregate in the bottom left, converging quickly to small loss values in most cases. The right column plots the log heldout loss by iteration corresponding to the first train-test split. The black dotted line corresponds to the multiplicative updates. The multiplicative updates decrease the loss much more rapidly than its competitors. The remaining $(\alpha, \beta)$ parameterizations and numerical results corresponding to ICEWS and WITS are shown in Appendix C, where these patterns consistently hold.

**Model comparison.** Figure 3 shows the custom models attain lower heldout loss values than their baselines.

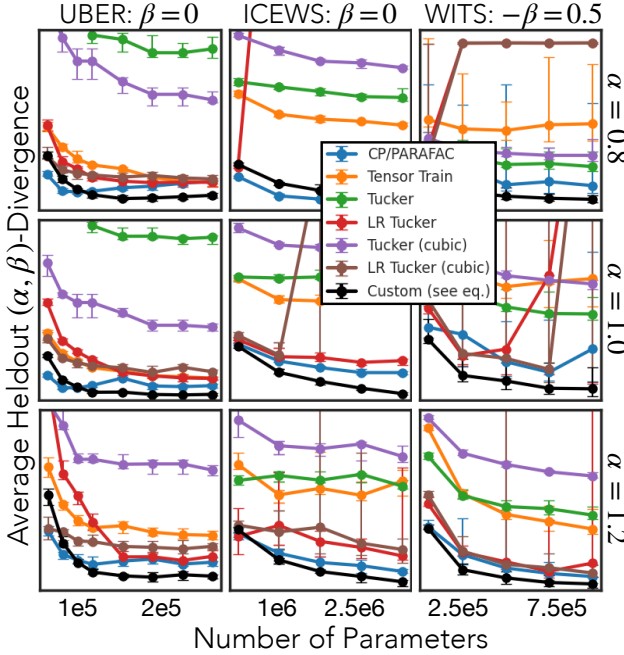

*Figure 3.* **Custom models attain lower heldout loss than common ones.** Model comparison: fit, as measured by average heldout loss, to three large tensor datasets (Uber, ICEWS, WITS). Error bars represent the interquartile range across random train-test splits. For each dataset, a different custom model achieves the lowest heldout loss. Each loss is tailored to the data. We set $\beta = 0$ for count data and $\beta = -0.5$ for sparse, positive continuous data. $\alpha$ controls a model's robustness to outliers and model misspecification.

The only exception occurs for ICEWS, $\alpha = 0.8$, where CP offers a 1% reduction in heldout loss to the custom model. The CP and Tucker instantiations correspond to the multiplicative update method, which outperformed their respective HALS implementations. This is not surprising, since Tucker-HALS and CP-HALS minimize a different objective than the other methods. At times, LR Tucker (red, brown) quickly converges to a poor stationary point ($\alpha = 0.8, 1.0$ and ICEWS, WITS). When avoiding this situation, LR Tucker often attains much lower loss values than its full version (green, purple). Overall, these results highlight how NNEinFact's ability to fit custom models offers empirical improvement over existing models.

**NNEinFact recovers interpretable qualitative structure.**
Finally, we highlight the interpretable qualitative structure uncovered by the custom model applied to the Uber data. We further partition the $100 \times 100$ spatial grid into a $400 \times 400$ mesh, set the number of temporal classes to $R = 10$ and the number of temporal-specific spatial factors to $K = 6$. Three of these classes are shown in Figure 4. With only $48,580$ parameters (relative to the 725 million entries) this model recovers classes corresponding to interpretable spatiotemporal structure. The first row of Figure 4 captures weekday morning rush-hour rides across Manhattan. The second shows weekday afternoon commutes originating in Midtown Manhattan, the city's primary central business district. The third reflects weekend nightlife concentrated in and around the West Village. Capturing similarly rich spatial structure with standard models such as CP would require either a large rank ($R \gg 10$), sacrificing parsimonious temporal structure, or coupling latitude and longitude into a $400^2$-dimensional spatial mode, prohibitively increasing the required number of parameters.

# 7. Discussion and Conclusion

NNEinFact is most valuable when exploring custom factorizations beyond CP and Tucker, using loss functions other than least-squares, or rapidly prototyping tensor models. Like most nonnegative algorithms, NNEinFact likely benefits from multiple random initializations. For standard CP or Tucker with least-squares, specialized algorithms such as HALS may offer empirical advantages. There are many future directions of this work.

One future direction focuses on scaling NNEinFact to work in settings where the data not fit into memory. Extensions may draw from the generalized CP (GCP) decomposition (Hong et al., 2020; Kolda & Hong, 2020). Like NNEinFact, GCP allows for a wide range of loss functions to accommodate both count and binary-valued data, among others. While GCP only considers the CP factorization structure, it uses stochastic gradient-based methods for large-scale optimization, and adapting ideas

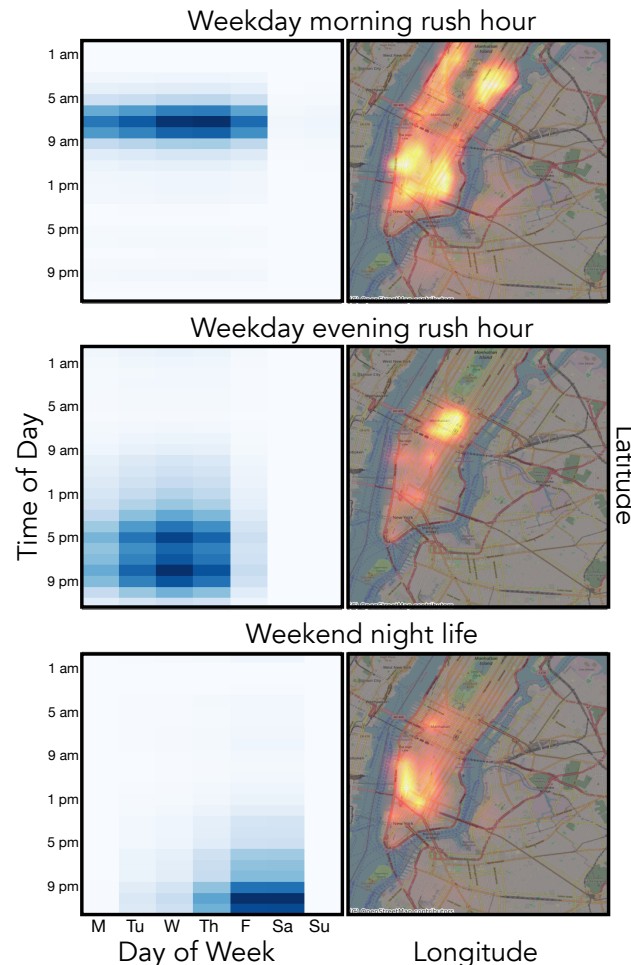

*Figure 4.* **NNEinFact uncovers interpretable spatiotemporal structure.** Each row depicts a latent class $r$ from the model `wr,dr,hr,irk,jrk → wdhij`. The left column shows temporal patterns by time of day and day of week, while the right column shows spatial loadings. The first two rows capture weekday morning and evening commutes, and the third captures weekend nightlife. Combined with the spatial loadings, these classes reveal interpretable spatiotemporal structure: morning commutes originate in residential areas, evening commutes in Midtown Manhattan, and nightlife in and around the West Village.

from this line of work may prove useful.

NNEinFact can serve as a foundation for modern tensor methods as multiway data becomes increasingly prevalent. The connection between ($\alpha$, $\beta$)-divergences and probabilistic models motivates principled approaches to scientific modeling. The computational efficiency of einsum-based updates, combined with rapidly improving hardware, may enable greater scaling. Beyond nonnegative tensor decomposition, NNEinFact may prove valuable in areas of machine learning where tensor methods are becoming increasingly relevant, such as tensor-on-tensor regression (Lock, 2018; Llosa-Vite & Maitra, 2022) and probabilistic circuits (Loconte et al., 2025).

## Acknowledgements

We would like to thank Ali Taylan Cemgil, Jie Jian, Jimmy Lederman, Nicolas Gillis, and Tammy Kolda for helpful discussions and feedback. JH is supported by the National Science Foundation under Grant No. 2140001. Part of the computing for this project was conducted on the University of Chicago's Data Science Institute cluster.

## Impact Statement

This paper presents work whose goal is to advance the field of Machine Learning. There are many potential societal consequences of our work, none which we feel must be specifically highlighted here.

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

# A. Proofs of Theoretical Results

**Proof of Lemma 4.1.**

**Construction of the surrogate function $Q$.** First we show $Q(\widetilde{\Theta} \mid \widetilde{\Theta}) = \mathcal{L}(\widetilde{\Theta})$. By definition,

$$Q(\widetilde{\Theta} \mid \widetilde{\Theta}) = \sum_{\mathbf{i}} \sum_{\mathbf{r}_\ell} \frac{\tilde{y}_{\mathbf{i},\mathbf{r}_\ell}}{\tilde{y}_{\mathbf{i}}} \mathcal{L}^{\text{vex}}(y_{\mathbf{i}}, \tilde{y}_{\mathbf{i}} \frac{\tilde{\theta}^{(\ell)}_{\mathbf{i}_\ell,\mathbf{r}_\ell}}{\tilde{\theta}^{(\ell)}_{\mathbf{i}_\ell,\mathbf{r}_\ell}}) + \mathcal{L}^{\text{cave}}(y_{\mathbf{i}}, \tilde{y}_{\mathbf{i}}) + \partial_y\left[\mathcal{L}^{\text{cave}}(y_{\mathbf{i}}, \tilde{y}_{\mathbf{i}})\right] \underbrace{(\tilde{y}_{\mathbf{i}} - \tilde{y}_{\mathbf{i}})}_{0} \tag{22}$$

$$= \sum_{\mathbf{i}} \mathcal{L}^{\text{vex}}(y_{\mathbf{i}}, \tilde{y}_{\mathbf{i}}) \underbrace{\sum_{\mathbf{r}_\ell} \frac{\tilde{y}_{\mathbf{i},\mathbf{r}_\ell}}{\tilde{y}_{\mathbf{i}}}}_{1} + \mathcal{L}^{\text{cave}}(y_{\mathbf{i}}, \tilde{y}_{\mathbf{i}}) \tag{23}$$

$$= \sum_{\mathbf{i}} \mathcal{L}(y_{\mathbf{i}}, \tilde{y}_{\mathbf{i}}) = \mathcal{L}(\mathcal{Y}, \widetilde{\mathcal{Y}}) = \mathcal{L}(\widetilde{\Theta}). \tag{24}$$

We apply Jensen's inequality to bound the convex component and a first-order Taylor approximation to bound the concave component.

**Bounding the convex component.** Letting $\hat{y}_{\mathbf{i},\mathbf{r}} = \prod_{\ell=1}^{L} \theta_{\mathbf{i}_\ell,\mathbf{r}_\ell}$, we wish to show that

$$\mathcal{L}^{\text{vex}}(y_{\mathbf{i}}, \hat{y}_{\mathbf{i}}) \leq \sum_{\mathbf{r}_\ell} \frac{\tilde{y}_{\mathbf{i},\mathbf{r}_\ell}}{\tilde{y}_{\mathbf{i}}} \mathcal{L}^{\text{vex}}(y_{\mathbf{i}}, \tilde{y}_{\mathbf{i}} \frac{\theta^{(\ell)}_{\mathbf{i}_\ell,\mathbf{r}_\ell}}{\tilde{\theta}^{(\ell)}_{\mathbf{i}_\ell,\mathbf{r}_\ell}}). \tag{25}$$

First, note that $\frac{\hat{y}_{\mathbf{i},\mathbf{r}_\ell}}{\tilde{y}_{\mathbf{i},\mathbf{r}_\ell}} = \frac{\theta_{\mathbf{i}_\ell,\mathbf{r}_\ell}}{\tilde{\theta}_{\mathbf{i}_\ell,\mathbf{r}_\ell}}$, and so we can write

$$\sum_{\mathbf{r}_\ell} \frac{\tilde{y}_{\mathbf{i},\mathbf{r}_\ell}}{\tilde{y}_{\mathbf{i}}} \mathcal{L}^{\text{vex}}(y_{\mathbf{i}}, \tilde{y}_{\mathbf{i}} \frac{\theta^{(\ell)}_{\mathbf{i}_\ell,\mathbf{r}_\ell}}{\tilde{\theta}^{(\ell)}_{\mathbf{i}_\ell,\mathbf{r}_\ell}}) = \sum_{\mathbf{r}_\ell} \frac{\tilde{y}_{\mathbf{i},\mathbf{r}_\ell}}{\tilde{y}_{\mathbf{i}}} \mathcal{L}^{\text{vex}}(y_{\mathbf{i}}, \tilde{y}_{\mathbf{i}} \frac{\hat{y}_{\mathbf{i},\mathbf{r}_\ell}}{\tilde{y}_{\mathbf{i},\mathbf{r}_\ell}}) \tag{26}$$

$$= \sum_{\mathbf{r}_\ell} \frac{\tilde{y}_{\mathbf{i},\mathbf{r}_\ell}}{\tilde{y}_{\mathbf{i}}} \mathcal{L}^{\text{vex}}(y_{\mathbf{i}}, \hat{y}_{\mathbf{i},\mathbf{r}_\ell} \frac{\tilde{y}_{\mathbf{i}}}{\tilde{y}_{\mathbf{i},\mathbf{r}_\ell}}). \tag{27}$$

Since $\sum_{\mathbf{r}_\ell} \frac{\tilde{y}_{\mathbf{i},\mathbf{r}_\ell}}{\tilde{y}_{\mathbf{i}}} = \frac{\tilde{y}_{\mathbf{i}}}{\tilde{y}_{\mathbf{i}}} = 1$, Jensen's inequality implies that

$$\sum_{\mathbf{r}_\ell} \frac{\tilde{y}_{\mathbf{i},\mathbf{r}_\ell}}{\tilde{y}_{\mathbf{i}}} \mathcal{L}^{\text{vex}}(y_{\mathbf{i}}, \hat{y}_{\mathbf{i},\mathbf{r}_\ell} \frac{\tilde{y}_{\mathbf{i}}}{\tilde{y}_{\mathbf{i},\mathbf{r}_\ell}}) \geq \mathcal{L}^{\text{vex}}(y_{\mathbf{i}}, \sum_{\mathbf{r}_\ell} \frac{\tilde{y}_{\mathbf{i},\mathbf{r}_\ell}}{\tilde{y}_{\mathbf{i}}} \left( \hat{y}_{\mathbf{i},\mathbf{r}_\ell} \frac{\tilde{y}_{\mathbf{i}}}{\tilde{y}_{\mathbf{i},\mathbf{r}_\ell}} \right)) \tag{28}$$

$$= \mathcal{L}^{\text{vex}}(y_{\mathbf{i}}, \sum_{\mathbf{r}_\ell} \hat{y}_{\mathbf{i},\mathbf{r}_\ell}) = \mathcal{L}^{\text{vex}}(y_{\mathbf{i}}, \hat{y}_{\mathbf{i}}) \tag{29}$$

where (29) follows from Jensen's inequality.

**Bounding the concave component.** For any concave function $f$, the first-order Taylor expansion of $f$ at $\widetilde{\Theta}$ yields the inequality $f(\Theta) \leq f(\widetilde{\Theta}) + \nabla f(\widetilde{\Theta})^\top (\Theta - \widetilde{\Theta})$. We leverage this property of $\mathcal{L}^{\text{cave}}(y_{\mathbf{i}}, \hat{y}_{\mathbf{i}})$ to write

$$\mathcal{L}^{\text{cave}}(y_{\mathbf{i}}, \hat{y}_{\mathbf{i}}) \leq \mathcal{L}^{\text{cave}}(y_{\mathbf{i}}, \tilde{y}_{\mathbf{i}}) + \partial_y \mathcal{L}^{\text{cave}}(y_{\mathbf{i}}, \tilde{y}_{\mathbf{i}})(\hat{y}_{\mathbf{i}} - \tilde{y}_{\mathbf{i}}). \tag{30}$$

Putting these parts together,

$$Q(\Theta \mid \widetilde{\Theta}) = \sum_{\mathbf{i}} \sum_{\mathbf{r}_\ell} \frac{\tilde{y}_{\mathbf{i},\mathbf{r}_\ell}}{\tilde{y}_{\mathbf{i}}} \mathcal{L}^{\text{vex}}(y_{\mathbf{i}}, \tilde{y}_{\mathbf{i}} \frac{\theta^{(\ell)}_{\mathbf{i}_\ell,\mathbf{r}_\ell}}{\tilde{\theta}^{(\ell)}_{\mathbf{i}_\ell,\mathbf{r}_\ell}}) + \mathcal{L}^{\text{cave}}(y_{\mathbf{i}}, \tilde{y}_{\mathbf{i}}) + \partial_y \mathcal{L}^{\text{cave}}(y_{\mathbf{i}}, \tilde{y}_{\mathbf{i}})(\hat{y}_{\mathbf{i}} - \tilde{y}_{\mathbf{i}}) \tag{31}$$

$$\geq \sum_{\mathbf{i}} \mathcal{L}^{\text{vex}}(y_{\mathbf{i}}, \hat{y}_{\mathbf{i}}) + \mathcal{L}^{\text{cave}}(y_{\mathbf{i}}, \hat{y}_{\mathbf{i}}) \tag{32}$$

$$= \sum_{\mathbf{i}} \mathcal{L}(y_{\mathbf{i}}, \hat{y}_{\mathbf{i}}) = \mathcal{L}(\Theta). \tag{33}$$

**Proof of Lemma 4.2.**

**Convexity of $Q$.** The Taylor expansion term

$$\mathcal{L}^{\text{cave}}(y_{\mathbf{i}}, \tilde{y}_{\mathbf{i}}) + \partial_y \mathcal{L}^{\text{cave}}(y_{\mathbf{i}}, \tilde{y}_{\mathbf{i}})(\hat{y}_{\mathbf{i}} - \tilde{y}_{\mathbf{i}}) \tag{34}$$

is linear in $\Theta$ and is thus convex. Consider the term

$$\sum_{\mathbf{r}_\ell} \frac{\tilde{y}_{\mathbf{i},\mathbf{r}_\ell}}{\tilde{y}_{\mathbf{i}}} \mathcal{L}^{\text{vex}}(y_{\mathbf{i}}, \tilde{y}_{\mathbf{i}} \frac{\theta^{(\ell)}_{\mathbf{i}_\ell,\mathbf{r}_\ell}}{\tilde{\theta}^{(\ell)}_{\mathbf{i}_\ell,\mathbf{r}_\ell}}). \tag{35}$$

It is a weighted sum of terms $\mathcal{L}^{\text{vex}}(y_{\mathbf{i}}, \tilde{y}_{\mathbf{i}} \frac{\theta^{(\ell)}_{\mathbf{i}_\ell,\mathbf{r}_\ell}}{\tilde{\theta}^{(\ell)}_{\mathbf{i}_\ell,\mathbf{r}_\ell}})$, each of which is convex in its second argument. Since affine transformations and nonnegative weighted sums of convex functions preserve convexity, this term is convex.

Again, since sums of convex terms are convex, it holds that $Q$ is convex in $\Theta$.

**Minimizing $Q$.** $Q(\Theta \mid \widetilde{\Theta})$ is proportional in $\Theta$ to

$$Q(\Theta \mid \widetilde{\Theta}) \propto_\Theta \sum_{\mathbf{i}} \sum_{\mathbf{r}_\ell} \frac{\tilde{y}_{\mathbf{i},\mathbf{r}_\ell}}{\tilde{y}_{\mathbf{i}}} \mathcal{L}^{\text{vex}}(y_{\mathbf{i}}, \tilde{y}_{\mathbf{i}} \frac{\theta^{(\ell)}_{\mathbf{i}_\ell,\mathbf{r}_\ell}}{\tilde{\theta}^{(\ell)}_{\mathbf{i}_\ell,\mathbf{r}_\ell}}) + \partial_y \mathcal{L}^{\text{cave}}(y_{\mathbf{i}}, \tilde{y}_{\mathbf{i}})\hat{y}_{\mathbf{i}} \tag{36}$$

with gradient given element-wise as

$$\frac{\partial}{\partial \theta_{\mathbf{i}_\ell,\mathbf{r}_\ell}} \left[ Q(\Theta \mid \widetilde{\Theta}) \right] = \sum_{\mathbf{i}} \frac{\tilde{y}_{\mathbf{i},\mathbf{r}_\ell}}{\tilde{y}_{\mathbf{i}}} \frac{\tilde{y}_{\mathbf{i}}}{\tilde{\theta}_{\mathbf{i}_\ell,\mathbf{r}_\ell}} \partial_y \mathcal{L}^{\text{vex}}(y_{\mathbf{i}}, \tilde{y}_{\mathbf{i}} \frac{\theta^{(\ell)}_{\mathbf{i}_\ell,\mathbf{r}_\ell}}{\tilde{\theta}^{(\ell)}_{\mathbf{i}_\ell,\mathbf{r}_\ell}}) + \partial_y \mathcal{L}^{\text{cave}}(y_{\mathbf{i}}, \tilde{y}_{\mathbf{i}}) \frac{\partial \hat{y}_{\mathbf{i}}}{\partial \theta_{\mathbf{i}_\ell,\mathbf{r}_\ell}} \tag{37}$$

$$= \sum_{\mathbf{i}} \frac{\partial \hat{y}_{\mathbf{i}}}{\partial \theta_{\mathbf{i}_\ell,\mathbf{r}_\ell}} \left( \partial_y \mathcal{L}^{\text{vex}}(y_{\mathbf{i}}, \tilde{y}_{\mathbf{i}} \frac{\theta^{(\ell)}_{\mathbf{i}_\ell,\mathbf{r}_\ell}}{\tilde{\theta}^{(\ell)}_{\mathbf{i}_\ell,\mathbf{r}_\ell}}) + \partial_y \mathcal{L}^{\text{cave}}(y_{\mathbf{i}}, \tilde{y}_{\mathbf{i}}) \right). \tag{38}$$

Letting $\lambda = \frac{\theta^{(\ell)}_{\mathbf{i}_\ell,\mathbf{r}_\ell}}{\tilde{\theta}^{(\ell)}_{\mathbf{i}_\ell,\mathbf{r}_\ell}}$ and setting $\frac{\partial}{\partial \theta_{\mathbf{i}_\ell,\mathbf{r}_\ell}} Q(\Theta \mid \widetilde{\Theta}) = 0$, we have that

$$\sum_{\mathbf{i}} \frac{\partial \hat{y}_{\mathbf{i}}}{\partial \theta_{\mathbf{i}_\ell,\mathbf{r}_\ell}} (\partial_y \mathcal{L}^{\text{vex}}(y_{\mathbf{i}}, \tilde{y}_{\mathbf{i}}\lambda) + \partial_y \mathcal{L}^{\text{cave}}(y_{\mathbf{i}}, \tilde{y}_{\mathbf{i}})) = 0. \tag{39}$$

Under the relation $\partial_y \mathcal{L}^{\text{vex}}(y_{\mathbf{i}}, \tilde{y}_{\mathbf{i}}\lambda) + \partial_y \mathcal{L}^{\text{cave}}(y_{\mathbf{i}}, \tilde{y}_{\mathbf{i}}) = [c(\lambda)(g(\lambda)b(y_{\mathbf{i}}, \hat{y}_{\mathbf{i}}) - a(y_{\mathbf{i}}, \hat{y}_{\mathbf{i}}))]$

$$c(\lambda)g(\lambda) \sum_{\mathbf{i}} \frac{\partial \hat{y}_{\mathbf{i}}}{\partial \theta_{\mathbf{i}_\ell,\mathbf{r}_\ell}} b(y_{\mathbf{i}}, \hat{y}_{\mathbf{i}}) = c(\lambda) \sum_{\mathbf{i}} \frac{\partial \hat{y}_{\mathbf{i}}}{\partial \theta_{\mathbf{i}_\ell,\mathbf{r}_\ell}} a(y_{\mathbf{i}}, \hat{y}_{\mathbf{i}}) \tag{40}$$

which implies that

$$\lambda = \frac{\theta^{(\ell)}_{\mathbf{i}_\ell,\mathbf{r}_\ell}}{\tilde{\theta}^{(\ell)}_{\mathbf{i}_\ell,\mathbf{r}_\ell}} = g^{-1} \left( \frac{\sum_{\mathbf{i}} \frac{\partial \hat{y}_{\mathbf{i}}}{\partial \theta_{\mathbf{i}_\ell,\mathbf{r}_\ell}} a(y_{\mathbf{i}}, \hat{y}_{\mathbf{i}})}{\sum_{\mathbf{i}} \frac{\partial \hat{y}_{\mathbf{i}}}{\partial \theta_{\mathbf{i}_\ell,\mathbf{r}_\ell}} b(y_{\mathbf{i}}, \hat{y}_{\mathbf{i}})} \right). \tag{41}$$

Multiplying both sides by $\tilde{\theta}^{(\ell)}_{\mathbf{i}_\ell,\mathbf{r}_\ell}$ yields the multiplicative update. If the multiplicative update lies outside of $[\epsilon, \infty)$, the minimum is attained at $\theta^{(\ell)}_{\mathbf{i}_\ell,\mathbf{r}_\ell} = \epsilon$ due to the convexity of $Q$.

**Proof of convergence to a stationary point.** We draw from the work of Razaviyayn et al. (2013), Theorem 2, which establishes convergence to stationary points for a class of algorithms referred to as *Block Successive Upper-bound Minimization Algorithms*. Algorithm 1 is one of these. In particular,

- $Q$ is quasi-convex in $\Theta^{(\ell)}$.

- $Q(\Theta^{(\ell)} \mid \widetilde{\Theta}^{(\ell)})$ has a unique minimum in $\Theta^{(\ell)}$.

- $Q(\Theta^{(\ell)} \mid \Theta^{(\ell)}) = \mathcal{L}(\Theta^{(\ell)})$ for all $\Theta^{(\ell)} \geq \epsilon$.

- $Q(\Theta^{(\ell)} \mid \tilde{\Theta}^{(\ell)}) \geq \mathcal{L}(\Theta^{(\ell)})$ for all $\Theta^{(\ell)}, \tilde{\Theta}^{(\ell)} \geq \epsilon$.

- $\nabla_{\Theta^{(\ell)}} Q(\Theta^{(\ell)} \mid \tilde{\Theta}^{(\ell)}) = \nabla_{\Theta^{(\ell)}} \mathcal{L}(\Theta^{(\ell)})$ at $\tilde{\Theta}^{(\ell)} = \Theta^{(\ell)}$ for all $\Theta^{(\ell)} \geq \epsilon$.

- $Q(\Theta^{(\ell')} \mid \tilde{\Theta}^{(\ell')})$ is continuous in $\{\Theta^{(\ell)}\}_{\ell=1}^{L}$ for all $\Theta^{(\ell)} \geq \epsilon, \ell \in [L]$.

The convexity of $Q(\Theta^{(\ell)} \mid \tilde{\Theta}^{(\ell)})$ in $\Theta^{(\ell)}$ implies that $Q$ is quasi-convex (Gillis, 2020). The closed-form expression for the minimum of $Q(\Theta^{(\ell)} \mid \tilde{\Theta}^{(\ell)})$ implies uniqueness, while the third and fourth statements are the surrogate property. A quick evaluation of (38) at $\tilde{\Theta}^{(\ell)} = \Theta^{(\ell)}$ implies tangency and the continuity of $Q$ follows from construction and the continuity of $\mathcal{L}$.

Since these statements hold, Theorem 2 of Razaviyayn et al. (2013) applies: every limit point is a stationary point of objective 8.

**Extension to regularized objectives.** Given a loss function $\mathcal{L}$ satisfying decomposability property (10), we consider penalized objectives of the form

$$\mathcal{L}(\mathcal{Y}, \hat{\mathcal{Y}}) + \sum_{\ell=1}^{L} R^{(\ell)}(\Theta^{(\ell)}), \quad \Theta^{(\ell)} \geq \epsilon \tag{42}$$

for differentiable $R^{(\ell)}$. The corresponding surrogate function is given by

$$\tilde{Q}(\Theta^{(\ell)} \mid \tilde{\Theta}^{(\ell)}) = Q(\Theta^{(\ell)} \mid \tilde{\Theta}^{(\ell)}) + \sum_{\ell=1}^{L} R^{(\ell)}(\Theta^{(\ell)}), \quad \Theta^{(\ell)} \geq \epsilon. \tag{43}$$

Minimizing $\tilde{Q}(\Theta^{(\ell)} \mid \tilde{\Theta}^{(\ell)})$ entails setting the gradient to zero and solving for $\Theta^{(\ell)}$, i.e.

$$\nabla_{\Theta^{(\ell)}} Q(\Theta^{(\ell)} \mid \tilde{\Theta}^{(\ell)}) + \nabla_{\Theta^{(\ell)}} R(\Theta^{(\ell)}) = 0, \tag{44}$$

where the form of $\nabla_{\Theta^{(\ell)}} Q(\Theta^{(\ell)} \mid \tilde{\Theta}^{(\ell)})$ is given in the proof of Lemma 4.2. Dropping the dependence on $\ell$ without a loss of generality, we can write

$$\nabla_{\Theta} R(\Theta) = \nabla_{\Theta} R\left(\Theta \frac{\tilde{\Theta}}{\tilde{\Theta}}\right) = \nabla_{\Theta} R\left(\frac{\Theta}{\tilde{\Theta}} \tilde{\Theta}\right) = \nabla_{\Theta} R(\lambda \tilde{\Theta}). \tag{45}$$

Now, if we can write $\nabla_{\Theta} R(\lambda \tilde{\Theta}) = c(\lambda)[g(\lambda)\tilde{b}(\tilde{\Theta}) - \tilde{a}(\tilde{\Theta})]$, then the corresponding multiplicative update is given by

$$\Theta^{(\ell)} \leftarrow \min\left(\epsilon, \tilde{\Theta}^{(\ell)} \odot g^{-1}\left(\frac{\tilde{a}(\tilde{\Theta}^{(\ell)}) + \sum_{\mathbf{i}}[\nabla_{\Theta^{(\ell)}} \hat{y}_{\mathbf{i}}] a(y_{\mathbf{i}}, \hat{y}_{\mathbf{i}})}{\tilde{b}(\tilde{\Theta}^{(\ell)}) + \sum_{\mathbf{i}}[\nabla_{\Theta^{(\ell)}} \hat{y}_{\mathbf{i}}] b(y_{\mathbf{i}}, \hat{y}_{\mathbf{i}})}\right)\right). \tag{46}$$

While this condition may not be trivial, it is satisfied in many settings, including $l_2$ loss, $l_1$ and $l_2$ regularization, KL-divergence loss and $l_1$ regularization. Settings where the loss is the negative log-likelihood of an exponential family distribution and the penalty corresponds to its conjugate prior also often suffice, such as in the Poisson-gamma setting.

**Some examples.** For $\mathcal{L}(x, y) = \frac{1}{2}(x - y)^2$, $g(\lambda) = \lambda$, $a(x, y) = x$, $b(x, y) = y$, $c(\lambda) = 1$. The $\ell_2$ regularization decomposes as

$$R(\Theta^{(\ell)}) = \frac{\gamma}{2} \sum_{\mathbf{i}_\ell, \mathbf{r}_\ell} (\theta_{\mathbf{i}_\ell, \mathbf{r}_\ell}^{(\ell)})^2, \quad \nabla_{\Theta^{(\ell)}} R(\Theta^{(\ell)}) = \gamma \Theta^{(\ell)} = \gamma \lambda \tilde{\Theta}^{(\ell)} = g(\lambda) \underbrace{\gamma \tilde{\Theta}^{(\ell)}}_{\tilde{b}(\tilde{\Theta}^{(\ell)})}, \tag{47}$$

and so the multiplicative update is

$$\Theta^{(\ell)} \leftarrow \min\left(\epsilon, \tilde{\Theta}^{(\ell)} \odot \frac{\sum_{\mathbf{i}}[\nabla_{\Theta^{(\ell)}} \hat{y}_{\mathbf{i}}] y_{\mathbf{i}}}{\gamma \tilde{\Theta}^{(\ell)} + \sum_{\mathbf{i}}[\nabla_{\Theta^{(\ell)}} \hat{y}_{\mathbf{i}}] \hat{y}_{\mathbf{i}}}\right). \tag{48}$$

For the standard $\ell_1$ penalty, $R(\Theta^{(\ell)}) = \gamma \sum_{\mathbf{i}_\ell, \mathbf{r}_\ell} \theta_{\mathbf{i}_\ell, \mathbf{r}_\ell}^{(\ell)}$, a little bit of algebra reveals that $\tilde{b}(\widetilde{\Theta}^{(\ell)}) = 0$ and $\tilde{a}(\widetilde{\Theta}^{(\ell)}) = -\gamma$. As such, the multiplicative update is

$$\Theta^{(\ell)} \leftarrow \min\left(\epsilon, \widetilde{\Theta}^{(\ell)} \odot \frac{\sum_{\mathbf{i}}[\nabla_{\Theta^{(\ell)}}\hat{y}_{\mathbf{i}}]y_{\mathbf{i}} - \gamma\widetilde{\Theta}^{(\ell)}}{\sum_{\mathbf{i}}[\nabla_{\Theta^{(\ell)}}\hat{y}_{\mathbf{i}}]\hat{y}_{\mathbf{i}}}\right). \tag{49}$$

For the KL divergence and $\ell_1$ regularization, the multiplicative update is

$$\Theta^{(\ell)} \leftarrow \min\left(\epsilon, \widetilde{\Theta}^{(\ell)} \odot \frac{\sum_{\mathbf{i}}[\nabla_{\Theta^{(\ell)}}\hat{y}_{\mathbf{i}}]\frac{y_{\mathbf{i}}}{\hat{y}_{\mathbf{i}}}}{\gamma + \sum_{\mathbf{i}}[\nabla_{\Theta^{(\ell)}}]\hat{y}_{\mathbf{i}}}\right). \tag{50}$$

## B. Algorithmic Details

All experiments were run on one GPU and all algorithms were implemented in Pytorch.

**Stopping criterion.** We use a variety of stopping criterion to evaluate convergence, including increasing validation loss for $5$ consecutive iterations, a decrease in the training loss of less than $10^{-6}$, or $5000$ iterations of training.

### B.1. Divergences

We implemented the $(\alpha, \beta)$-divergence (Cichocki & Amari, 2010) setting of Algorithm 1, a family of divergences parameterized by $\alpha, \beta \in \mathbb{R}$.

The $(\alpha, \beta)$-divergence is defined as

$$\mathcal{L}_{\alpha,\beta}(x, y) = \begin{cases} \frac{1}{\alpha\beta}\left[\frac{\alpha}{\alpha+\beta}x^{\alpha+\beta} + \frac{\beta}{\alpha+\beta}y^{\alpha+\beta} - x^\alpha y^\beta\right] & \alpha, \beta, \alpha+\beta \neq 0 \\[2mm] \frac{1}{\alpha^2}\left[y^\alpha - x^\alpha + \alpha x^\alpha \log\frac{x}{y}\right] & \beta = 0, \ \alpha \neq 0 \\[2mm] \frac{1}{\alpha^2}\left[\frac{x^\alpha}{y^\alpha} - 1 + \alpha \log\frac{y}{x}\right] & \alpha = -\beta \neq 0 \\[2mm] \frac{1}{\beta^2}\left[\beta y^\beta \log\frac{y}{x} - y^\beta + x^\beta\right] & \alpha = 0, \ \beta \neq 0 \\[2mm] \frac{1}{2}\left(\log x - \log y\right)^2 & \alpha = \beta = 0 \end{cases} \tag{51}$$

Special cases include the $\alpha$-divergence (Amari, 2007), for $\alpha + \beta = 1$ and the $\beta$-divergence (Basu et al., 1998) for $\alpha = 1$. Included are the KL divergence ($\alpha = 1, \beta = 0$), reverse KL ($\alpha = 0, \beta = 1$), squared Euclidean distance ($\alpha = 1, \beta = 1$), Itakura-Saito divergence ($\alpha = 1, \beta = -1$), as well as the squared Hellinger distance ($\alpha = \beta = 0.5$), Neyman $\chi^2$ ($\alpha = -1, \beta = 2$) and Pearson $\chi^2$ divergences ($\alpha = 2, \beta = -1$).

When $\alpha \neq 0$, $a(x, y) = x^\alpha y^{\beta-1}$ and $b(x, y) = y^{\alpha+\beta-1}$. When $\alpha = 0$, $\beta = 1$, $a(x, y) = \log(x/y)$ and $b(x, y) = 1$. Otherwise, when $\alpha = 0$ and $\beta \neq 1$, we do not find a decomposition satisfying the decomposability property (10).

$g(\lambda)$ is defined by:

$$g(\lambda) = \begin{cases} \lambda^{1-\beta} & 1/\alpha - \beta/\alpha > 1 \\ \lambda^{\alpha+\beta-1} & 1/\alpha - \beta/\alpha < 0 \\ \log(\lambda) & \alpha = 0, \beta = 1 \\ \lambda^\alpha & 0 \leq 1/\alpha - \beta/\alpha \leq 1 \end{cases} \tag{52}$$

**Maximum likelihood under the negative binomial.** The negative binomial random variable $X$ with mean $y$ and dispersion parameter $\phi$ has probability mass function

$$p(x \mid y, \phi) = \frac{\Gamma(x + \phi)}{\Gamma(\phi)\Gamma(x + 1)}\left(\frac{\phi}{\phi + y}\right)^\phi \left(\frac{y}{\phi + y}\right)^x. \tag{53}$$

The terms in the negative log-likelihood proportional to $y$ are given by

$$\mathcal{L}(x, y) = (\phi + x)\log(\phi + y) - x\log(y), \tag{54}$$

which decomposes into convex and concave parts

$$\mathcal{L}^{\text{vex}}(x, y) = -x\log(y), \quad \mathcal{L}^{\text{cave}}(x, y) = (\phi + x)\log(\phi + y). \tag{55}$$

Then

$$\partial_y \mathcal{L}^{\text{vex}}(x, \lambda y) + \partial_y \mathcal{L}^{\text{cave}}(x, y) = -\lambda^{-1}\frac{x}{y} + \frac{\phi + x}{\phi + y} \tag{56}$$

$$= \lambda^{-1}\left(\lambda\frac{\phi + x}{\phi + y} - \frac{x}{y}\right). \tag{57}$$

Here, $c(\lambda) = \lambda^{-1}$, $g(\lambda) = \lambda$, $a(x, y) = \frac{x}{y}$ and $b(x, y) = \frac{\phi + x}{\phi + y}$. Suppose each $y_{\mathbf{i}} \sim \text{NegBinom}(\hat{y}_{\mathbf{i}}, \phi_{\mathbf{i}})$. Then for fixed $\phi_{\mathbf{i}}$, the multiplicative update that minimizes the negative log-likelihood is

$$\Theta^{(\ell)} \leftarrow \Theta^{(\ell)} \odot \left(\frac{\sum_{\mathbf{i}} [\nabla_{\Theta^{(\ell)}} \hat{y}_{\mathbf{i}}] \frac{y_{\mathbf{i}}}{\hat{y}_{\mathbf{i}}}}{\sum_{\mathbf{i}} [\nabla_{\Theta^{(\ell)}} \hat{y}_{\mathbf{i}}] \frac{y_{\mathbf{i}} + \phi_{\mathbf{i}}}{\hat{y}_{\mathbf{i}} + \phi_{\mathbf{i}}}}\right). \tag{58}$$

The geometric distribution arises when $\phi = 1$.

**Maximum likelihood under the Bernoulli.** The Bernoulli random variable $X \sim \text{Bern}(p)$ has probability mass function

$$p(x \mid p) = p^x(1 - p)^{1-x} \tag{59}$$

for $x \in \{0, 1\}$. Reparameterizing using the odds ratio $\mu := \frac{p}{1-p}$, the negative log likelihood simplifies to

$$\mathcal{L}(x, \mu) = \log(1 + \mu) - x\log(\mu) \tag{60}$$

which decomposes into convex and concave parts

$$\mathcal{L}^{\text{vex}}(x, \mu) = -x\log(\mu), \quad \mathcal{L}^{\text{cave}}(x, \mu) = \log(1 + \mu). \tag{61}$$

Then

$$\partial_y \mathcal{L}^{\text{vex}}(x, \lambda y) + \partial_y \mathcal{L}^{\text{cave}}(x, y) = -\lambda^{-1}\frac{x}{y} + \frac{1}{1 + y} \tag{62}$$

$$= \lambda^{-1}\left(\lambda\frac{1}{1 + y} - \frac{x}{y}\right). \tag{63}$$

Here, $c(\lambda) = \lambda^{-1}$, $g(\lambda) = \lambda$, $a(x, y) = \frac{x}{y}$ and $b(x, y) = \frac{1}{1+y}$. Under odds estimation using Equation (1), the multiplicative update that minimizes the negative log-likelihood is

$$\Theta^{(\ell)} \leftarrow \Theta^{(\ell)} \odot \left(\frac{\sum_{\mathbf{i}} [\nabla_{\Theta^{(\ell)}} \hat{y}_{\mathbf{i}}] \frac{y_{\mathbf{i}}}{\hat{y}_{\mathbf{i}}}}{\sum_{\mathbf{i}} [\nabla_{\Theta^{(\ell)}} \hat{y}_{\mathbf{i}}] \frac{1}{\hat{y}_{\mathbf{i}} + 1}}\right). \tag{64}$$

This framework extends to the binomial setting where $y_{\mathbf{i}} \sim \text{Binomial}(n_{\mathbf{i}}, p_{\mathbf{i}})$. For known number of trials $n_{\mathbf{i}}$, the corresponding update is

$$\Theta^{(\ell)} \leftarrow \Theta^{(\ell)} \odot \left(\frac{\sum_{\mathbf{i}} [\nabla_{\Theta^{(\ell)}} \hat{y}_{\mathbf{i}}] \frac{y_{\mathbf{i}}}{\hat{y}_{\mathbf{i}}}}{\sum_{\mathbf{i}} [\nabla_{\Theta^{(\ell)}} \hat{y}_{\mathbf{i}}] \frac{n_{\mathbf{i}}}{\hat{y}_{\mathbf{i}} + 1}}\right). \tag{65}$$

In each of these settings, $\hat{y}_{\mathbf{i}}$ is the estimated odds $\hat{y}_{\mathbf{i}} := \frac{\hat{p}_{\mathbf{i}}}{1 - \hat{p}_{\mathbf{i}}}$.

**Jensen-Shannon divergence.** The Jensen-Shannon divergence is defined as

$$\mathcal{L}(x,y) = \tfrac{1}{2}x[\log(x) - \log(\tfrac{x+y}{2})] + \tfrac{1}{2}y[\log(y) - \log(\tfrac{x+y}{2})] \tag{66}$$

$$= -\tfrac{x+y}{2}\log(\tfrac{x+y}{2}) + \tfrac{1}{2}[x\log(x) + y\log(y)]. \tag{67}$$

It has convex-concave decomposition

$$\mathcal{L}^{\text{vex}}(x,y) = \tfrac{1}{2}[x\log(x) + y\log(y)], \quad \mathcal{L}^{\text{cave}}(x,y) = -\tfrac{x+y}{2}\log(\tfrac{x+y}{2}). \tag{68}$$

Then

$$\partial_y \mathcal{L}^{\text{vex}}(x,\lambda y) + \partial_y \mathcal{L}^{\text{cave}}(x,y) = \tfrac{1}{2}(1 + \log(\lambda) + \log(y) - \log(\tfrac{x+y}{2}) - 1) \tag{69}$$

$$= \tfrac{1}{2}[\log(\lambda) - \log(\tfrac{x+y}{2y})]. \tag{70}$$

Then $a(x,y) = \log(\tfrac{x+y}{2y})$, $b(x,y) = 1$, $g(\lambda) = \log(\lambda)$, and $c(\lambda) = \tfrac{1}{2}$. The multiplicative update is

$$\Theta^{(\ell)} \leftarrow \Theta^{(\ell)} \odot \exp\left( \frac{\sum_{\mathbf{i}} [\nabla_{\Theta^{(\ell)}} \hat{y}_{\mathbf{i}}] \log(\tfrac{y_{\mathbf{i}} + \hat{y}_{\mathbf{i}}}{2\hat{y}_{\mathbf{i}}})}{\sum_{\mathbf{i}} [\nabla_{\Theta^{(\ell)}} \hat{y}_{\mathbf{i}}]} \right) \tag{71}$$

## C. Additional Empirical Results

Figure 5 shows additional results from the comparisons to gradient-based automatic differentiation corresponding to $\alpha \in \{0.7, 1.0, 1.3\}, \beta \in \{0.0, 1.0\}$ for the Uber (left two columns), ICEWS (middle columns), and WITS tensors (right).

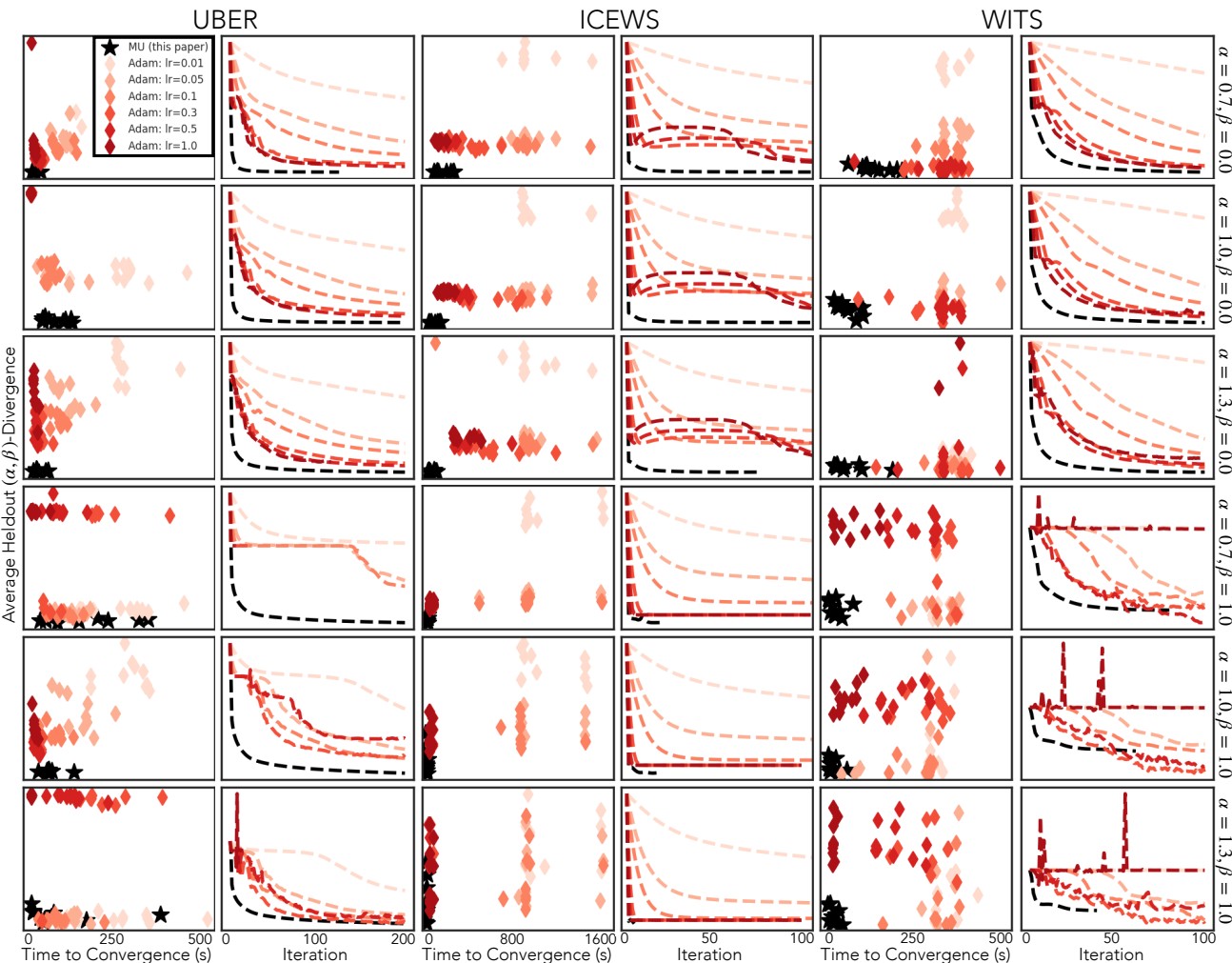

*Figure 5.* Side-by-side comparisons to automatic differentiation across a variety of $(\alpha, \beta)$ and datasets. NNEinFact's multiplicative update algorithm is shown in black, while the baselines take on different shades of red.

The per-iteration plots (right columns) show heldout loss on a logarithmic scale, while the time to convergence plots (left columns) show heldout loss on a linear scale. $\alpha = 1.0, \beta = 1.0$ corresponds to minimizing the squared Euclidean distance and $\alpha = 1.0, \beta = 0.0$ corresponds to minimizing the KL divergence.

*Table 1.* Mean heldout $(\alpha, \beta)$-divergences for NNEinFact and Adam. Lowest values in each row are bolded. Standard errors are shown in parentheses below each row.

| Dataset | $(\alpha, \beta)$ | NNEinFact | Adam, 0.01 | Adam, 0.05 | Adam, 0.1 | Adam, 0.3 | Adam, 0.5 | Adam, 1.0 |
|---------|-------------------|-----------|------------|------------|-----------|-----------|-----------|-----------|
| Uber | (0.7, 0) | **0.00759** | 0.0161 | 0.0131 | 0.0115 | 0.0100 | 0.00973 | 0.0127 |
| | | (0.00001) | (0.0004) | (0.0005) | (0.0002) | (0.0001) | (0.0001) | (0.0016) |
| | (0.7, 1) | **0.0444** | 0.1855 | 0.1695 | 0.1903 | 1.9391 | 1.9391 | 1.9412 |
| | | (0.0007) | (0.011) | (0.012) | (0.014) | (0.014) | (0.014) | (0.014) |
| | (1.0, 0) | **0.0108** | 0.0229 | 0.0190 | 0.0164 | 0.0149 | 0.0151 | 0.0195 |
| | | (0.00003) | (0.0006) | (0.0004) | (0.0002) | (0.0004) | (0.0004) | (0.0006) |
| | (1.0, 1) | **0.367** | 0.542 | 0.478 | 0.458 | 9.673 | 18.739 | 17.378 |
| | | (0.006) | (0.015) | (0.012) | (0.011) | (2) | (1.6) | (0.1) |
| | (1.3, 0) | **0.0225** | 0.0538 | 0.0418 | 0.0349 | 0.0323 | 0.0295 | 0.0366 |
| | | (0.0001) | (0.001) | (0.002) | (0.001) | (0.001) | (0.0005) | (0.001) |
| | (1.3, 1) | 1.032 | 0.939 | 0.917 | **0.896** | 34.099 | 40.061 | 42.121 |
| | | (0.06) | (0.03) | (0.03) | (0.02) | (3.7) | (0.9) | (0.4) |
| ICEWS | (0.7, 0) | **0.0185** | 0.0590 | 0.0305 | 0.0279 | 0.0281 | 0.0291 | 0.0292 |
| | | (0.00002) | (0.0007) | (0.0002) | (0.0002) | (0.0004) | (0.0002) | (0.0001) |
| | (0.7, 1) | **0.0926** | 0.5090 | 0.1867 | 0.1655 | 0.1523 | 0.1523 | 0.1523 |
| | | (0.003) | (0.013) | (0.003) | (0.003) | (0.004) | (0.004) | (0.004) |
| | (1.0, 0) | **0.0309** | 0.1178 | 0.0558 | 0.0503 | 0.0498 | 0.0532 | 0.0548 |
| | | (0.00004) | (0.002) | (0.001) | (0.001) | (0.001) | (0.001) | (0.001) |
| | (1.0, 1) | **0.7558** | 2.3336 | 1.2767 | 1.2055 | 1.1669 | 1.1671 | 1.1671 |
| | | (0.06) | (0.07) | (0.06) | (0.06) | (0.06) | (0.06) | (0.06) |
| | (1.3, 0) | **0.0699** | 0.2858 | 0.1370 | 0.1431 | 0.1135 | 0.1263 | 0.1390 |
| | | (0.0004) | (0.005) | (0.002) | (0.02) | (0.002) | (0.002) | (0.001) |
| | (1.3, 1) | **2.0350** | 4.0964 | 3.0809 | 3.0556 | 2.9887 | 2.9890 | 2.9890 |
| | | (0.2) | (0.3) | (0.3) | (0.3) | (0.3) | (0.3) | (0.3) |
| WITS | (0.7, 0) | **0.0086** | 0.0122 | 0.0099 | 0.0092 | **0.0086** | **0.0086** | 0.0196 |
| | | (0.00002) | (0.00007) | (0.00006) | (0.00004) | (0.00003) | (0.00004) | (0.0) |
| | (0.7, 1) | 0.0945 | **0.0736** | 0.1114 | 65.2589 | 0.2793 | 0.4506 | 0.4625 |
| | | (0.01) | (0.007) | (0.02) | (61.0) | (0.05) | (0.01) | (0.01) |
| | (1.0, 0) | 0.0132 | 0.0173 | 0.0140 | 0.0135 | 0.0132 | **0.0131** | nan |
| | | (0.00008) | (0.0001) | (0.00008) | (0.00006) | (0.0001) | (0.00007) | (nan) |
| | (1.0, 1) | **0.8026** | 1.9767 | 5.9140 | 6.9676 | 4.2028 | 4.4954 | 4.6227 |
| | | (0.1) | (0.7) | (4.0) | (6.0) | (0.2) | (0.2) | (0.2) |
| | (1.3, 0) | **0.0312** | 0.0422 | 0.0320 | 0.0314 | 0.0318 | 0.0444 | 0.1213 |
| | | (0.001) | (0.001) | (0.001) | (0.001) | (0.002) | (0.009) | (0.006) |
| | (1.3, 1) | **2.1723** | 240.4041 | 560.7473 | 143.2240 | 433.9093 | 13.5632 | 13.8273 |
| | | (0.4) | (210.0) | (340.0) | (120.0) | (400.0) | (0.8) | (0.9) |

*Table 2.* Mean runtime to convergence (in seconds) for NNEinFact and Adam baselines. We drop Adam, 1.0 from the comparison as it performs poorly relative to other baselines in Table 1. Lowest values in each row are bolded. Standard errors are shown in parentheses below each row.

| Dataset | $(\alpha, \beta)$ | NNEinFact | Adam, 0.01 | Adam, 0.05 | Adam, 0.1 | Adam, 0.3 | Adam, 0.5 |
|---|---|---|---|---|---|---|---|
| Uber | (0.7, 0) | **9.98** | 213.84 | 89.54 | 78.25 | 26.02 | 19.79 |
| | | (1.87) | (13.96) | (6.07) | (7.11) | (2.36) | (2.11) |
| | (0.7, 1) | 71.42 | 262.43 | 69.29 | 51.40 | 2.34 | **1.29** |
| | | (9.62) | (23.68) | (12.34) | (4.64) | (0.25) | (0.12) |
| | (1.0, 0) | 21.60 | 256.78 | 101.52 | 72.28 | 27.42 | **15.64** |
| | | (3.71) | (15.88) | (11.11) | (5.53) | (4.18) | (1.10) |
| | (1.0, 1) | 136.94 | 256.59 | 103.21 | 79.78 | 151.50 | **73.97** |
| | | (32.55) | (26.54) | (11.10) | (9.87) | (31.83) | (9.77) |
| | (1.3, 0) | 52.39 | 264.45 | 133.72 | 66.96 | 22.65 | **18.84** |
| | | (7.48) | (17.70) | (15.20) | (7.50) | (2.35) | (1.99) |
| | (1.3, 1) | 73.26 | 256.46 | 104.08 | **72.61** | 180.02 | 109.31 |
| | | (32.12) | (33.09) | (8.70) | (14.79) | (26.49) | (13.29) |
| ICEWS | (0.7, 0) | **150.34** | 905.15 | 873.73 | 883.17 | 373.39 | 179.04 |
| | | (19.45) | (63.21) | (29.86) | (59.37) | (41.83) | (14.08) |
| | (0.7, 1) | **17.39** | 1068.02 | 1012.34 | 953.73 | 60.05 | 56.68 |
| | | (2.11) | (90.93) | (106.03) | (94.33) | (4.78) | (4.75) |
| | (1.0, 0) | **97.24** | 1020.31 | 988.65 | 847.52 | 469.25 | 250.52 |
| | | (11.69) | (82.23) | (71.19) | (26.46) | (60.10) | (23.01) |
| | (1.0, 1) | **15.03** | 1060.25 | 982.92 | 904.38 | 55.08 | 50.22 |
| | | (1.77) | (78.49) | (77.31) | (73.65) | (4.26) | (3.85) |
| | (1.3, 0) | **57.32** | 924.56 | 920.01 | 849.07 | 550.83 | 363.19 |
| | | (7.92) | (63.12) | (57.74) | (96.13) | (74.36) | (43.81) |
| | (1.3, 1) | **11.97** | 1070.40 | 1043.53 | 909.01 | 49.60 | 44.79 |
| | | (1.64) | (87.95) | (94.82) | (66.34) | (4.20) | (3.40) |
| WITS | (0.7, 0) | **134.03** | 346.20 | 360.72 | 334.97 | 304.82 | 305.83 |
| | | (14.21) | (8.52) | (8.71) | (11.75) | (21.33) | (26.01) |
| | (0.7, 1) | **24.01** | 320.79 | 306.93 | 298.41 | 330.51 | 253.89 |
| | | (6.34) | (8.57) | (13.94) | (19.42) | (7.66) | (17.21) |
| | (1.0, 0) | **59.19** | 356.28 | 345.07 | 329.17 | 267.04 | 342.03 |
| | | (9.13) | (5.54) | (15.77) | (5.27) | (25.07) | (7.28) |
| | (1.0, 1) | **18.81** | 306.27 | 219.45 | 269.30 | 292.06 | 205.46 |
| | | (5.13) | (8.53) | (32.27) | (13.78) | (16.01) | (19.38) |
| | (1.3, 0) | **58.32** | 353.93 | 361.27 | 336.98 | 327.41 | 339.76 |
| | | (15.61) | (8.55) | (6.00) | (10.06) | (21.99) | (21.37) |
| | (1.3, 1) | **17.69** | 345.67 | 319.95 | 310.72 | 295.07 | 193.25 |
| | | (3.85) | (11.01) | (20.04) | (18.11) | (19.39) | (17.05) |

*Table 3.* Mean heldout $(\alpha, \beta)$-divergence for all models. We report the best values over different numbers of parameters. Lowest values in each row are bolded. Standard errors are shown in parentheses below each row. For CP and Tucker, we report values from the best-performing method (MU, HALS) for each instance.

| Dataset | $\alpha$ | Custom | CP/PARAFAC | Tensor-train | Tucker | LR Tucker | Tucker (cubic) | LR Tucker (cubic) |
|---------|----------|--------|-----------|--------------|--------|-----------|----------------|-------------------|
| Uber | 0.8 | **0.00804** | 0.00812 | 0.00823 | 0.00986 | 0.00823 | 0.00925 | 0.00828 |
| | | (0.00002) | (0.00002) | (0.00002) | (0.00004) | (0.00002) | (0.00004) | (0.00002) |
| | 1.0 | **0.0101** | 0.0104 | 0.0107 | 0.0149 | 0.0107 | 0.0123 | 0.0108 |
| | | (0.00003) | (0.00002) | (0.00004) | (0.00006) | (0.00003) | (0.00006) | (0.00007) |
| | 1.2 | **0.0152** | 0.0158 | 0.0169 | 0.0291 | 0.0160 | 0.0197 | 0.0164 |
| | | (0.00007) | (0.00013) | (0.00010) | (0.00016) | (0.00012) | (0.00015) | (0.00011) |
| ICEWS | 0.8 | 0.0203 | **0.0201** | 0.0226 | 0.0234 | 0.0212 | 0.0243 | 0.0354 |
| | | (0.00003) | (0.00001) | (0.00002) | (0.00006) | (0.00002) | (0.00004) | (0.00005) |
| | 1.0 | **0.0273** | 0.0285 | 0.0331 | 0.0342 | 0.0291 | 0.0356 | 0.0301 |
| | | (0.00012) | (0.00004) | (0.00012) | (0.00013) | (0.00005) | (0.00006) | (0.00031) |
| | 1.2 | **0.0457** | 0.0470 | 0.0560 | 0.0557 | 0.0595 | 0.0515 | 0.0515 |
| | | (0.0001) | (0.0001) | (0.001) | (0.0002) | (0.004) | (0.0004) | (0.002) |
| WITS | 0.8 | **0.166** | 0.234 | 0.361 | 0.269 | 0.313 | 0.246 | 0.297 |
| | | (0.004) | (0.018) | (0.038) | (0.031) | (0.032) | (0.008) | (0.023) |
| | 1.0 | **0.0356** | 0.0570 | 0.0780 | 0.0782 | 0.0742 | 0.0673 | 0.0739 |
| | | (0.002) | (0.005) | (0.011) | (0.012) | (0.011) | (0.007) | (0.005) |
| | 1.2 | **0.0134** | 0.0178 | 0.0294 | 0.0280 | 0.0293 | 0.0300 | 0.0246 |
| | | (0.0009) | (0.002) | (0.003) | (0.002) | (0.002) | (0.0003) | (0.004) |

**Uber qualitative comparison.** For comparison, we fit the CP decomposition with $R = 10$ classes to the Uber data. On the left, we show the classes recovered by the custom model corresponding to "weekday morning rush hour", "weekday evening rush hour", "weekend night life" in Figure 4. To the right, we show their most closely resembled classes recovered by CP. Not only does the custom model capture more complex spatial structure, the temporal structure is much more refined.

### C.1. Synthetic Experiments

We consider two synthetic settings to evaluate NNEinFact's ability to recover the latent underlying structure in settings where the ground truth is known.

**Factor matrix recovery.** The purpose of this study is to evaluate NNEinFact's recovery of underlying latent parameters. First, we select three non-traditional einsum factorization structures and generate data from them. To compare to CP and Tucker baselines, we consider settings with ground truth factor matrices. Therefore, each model (custom, which matches the true data-generating structure, CP, and Tucker) have factor matrices which match the structure of the ground truth.

Varying the sparsity level of the factors, we compute the cosine similarity of the estimated factors to the ground truth, repeating each experimental setting 10 times. The mean and standard errors are reported in the table below.

*Table 4.* The custom model recovers the ground truth factor matrices better than CP and Tucker uniformly across sparsity levels.

| % sparse | Custom | CP | Tucker |
|----------|--------|-----|--------|
| 0.3 | **0.99** (0.005) | 0.63 (0.003) | 0.93 (0.01) |
| 0.5 | **0.96** (0.01) | 0.59 (0.01) | 0.91 (0.02) |
| 0.7 | **0.92** (0.02) | 0.51 (0.01) | 0.83 (0.02) |
| 0.9 | **0.85** (0.02) | 0.50 (0.01) | 0.65 (0.04) |

**Tensor reconstruction.** We ran another set of synthetic experiments to evaluate NNEinfact's reconstruction capabilities when the ground truth is known and matches the fitted custom model. For 200 randomly-generated tensor network structures, we fit the custom model corresponding to the ground truth, as well as CP, Tucker, tensor train, tensor wheel, and tensor ring. The table below shows that NNEinfact consistently achieves better recovery, as measured by relative RMSE, than other tensor decomposition structures, including more modern ones (tensor train, tensor ring and tensor wheel), which fail to consistently recover the true underlying structure due to model misspecification. Empirically, the recovery discrepancy

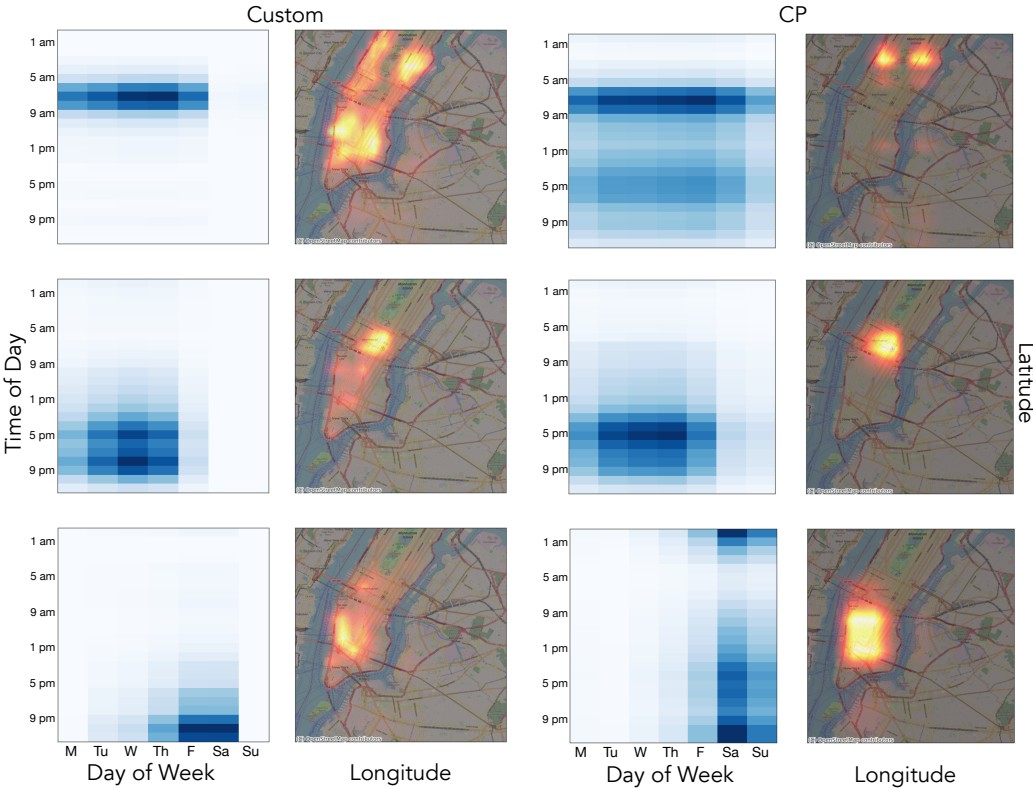

*Figure 6.* Qualitative side-by-side comparison of the custom (left) and CP (right) models.

increases with the number of modes.

*Table 5.* Mean reduction in relative RMSE (baseline - NNEinFact). Values greater than zero imply improvement over baseline.

| Model/modes | 3 | 4 | 5 | 6 | 7 | 8 |
|---|---|---|---|---|---|---|
| CP | −0.02 (0.01) | 0.05 (0.03) | 0.03 (0.03) | 0.10 (0.03) | 0.09 (0.05) | 0.20 (0.05) |
| Tucker | 0.04 (0.04) | 0.15 (0.04) | 0.18 (0.06) | 0.34 (0.08) | 0.68 (0.20) | 0.89 (0.15) |
| Tensor Train | 0.03 (0.06) | 0.19 (0.05) | 0.15 (0.06) | 0.37 (0.08) | 0.52 (0.09) | 0.91 (0.14) |
| Tensor Ring | 0.02 (0.03) | 0.14 (0.04) | 0.19 (0.06) | 0.30 (0.08) | 0.48 (0.07) | 0.94 (0.13) |
| Tensor Wheel | −0.01 (0.02) | 0.04 (0.02) | 0.08 (0.03) | 0.11 (0.03) | 0.35 (0.17) | 0.53 (0.12) |

We provide example model strings below:

4 modes: aP,bP,cQS,dRS,PQRS→ abcd
6 modes: aPQR,bPQS,cPRT,dQRT,ePST,fQST,PQRST → abcdef
8 modes: aPQR,bPQS,cPRT,dQRT,ePSU,fQTU,gRSV,hTUV,PQRSTUV → abcdefgh.

