# OpenReview forum: "Near-Universal Multiplicative Updates for Nonnegative Einsum Factorization"
_ICML.cc/2026/Conference — ICML 2026 regular_

### Official Review · Reviewer_zE2D · 2026-03-06

**Soundness:** 3
**Presentation:** 3
**Significance:** 3
**Originality:** 2
**Overall Recommendation:** 4
**Confidence:** 3

**Summary:**

This paper introduces NNEinfact (Nonnegative Einsum Factorization), a framework for nonnegative tensor factorization that unifies a broad family of decomposition models through einsum notation. The paper combines theoretical contributions (convex-concave decomposition of loss functions, derivation of multiplicative updates) with practical implementation and thorough empirical evaluation. The main contributions of the paper can be summarized as follows:

1. A unified formulation that expresses generalized tensor factorizations as einsum operations, allowing users to specify custom decomposition structures with minimal code.

2. Multiplicative update algorithms derived via majorization-minimization for the (α,β)-divergence family, with theoretical guarantees of monotonic convergence. The authors show these updates can be computed efficiently using einsum operations.

3. Broad applicability demonstrated on three real-world datasets (Uber, ICEWS, WITS) with custom models tailored to each domain's structure.

4. Empirical validation showing that the proposed multiplicative updates consistently outperform gradient-based methods (Adam) in both final held-out loss and convergence speed and that domain-informed custom models achieve lower loss than standard decompositions (CP, Tucker, tensor-train) with comparable parameter counts.

**Compliance With Llm Reviewing Policy:**

Affirmed.

**Final Justification:**

The authors addressed my concerns.

**Key Questions For Authors:**

1- Theoretical results guarantee monotonic decrease of the loss, but what can you say about the convergence rate? Is it linear, sublinear, or problem-dependent?

2- In practice, how should practitioners choose α and β for a new dataset? You mention connections to exponential families (β=0 for Poisson, β=-0.5 for compound Poisson-gamma), but is there guidance when the data-generating process is unknown? Could cross-validation be used effectively?

**Limitations:**

1- The requirement that loss functions decompose into convex and concave components (Property 1) excludes some potentially useful divergences. The authors acknowledge this and provide examples that satisfy it.

2- The multiplicative updates require computing gradients via einsum, which may still be expensive for very large tensors, though the authors demonstrate scalability to real-world datasets.

3- The custom model design relies on domain expertise, which may not always be available; the paper does not provide guidance on automated structure discovery.

**Strengths And Weaknesses:**

Strengths:

1- The theoretical derivation is rigorous and well-structured. The authors carefully establish the conditions under which loss functions can be decomposed into convex and concave components, then prove that their surrogate function Q satisfies the majorization-minimization conditions and yields closed-form multiplicative updates.

2- The connection to exponential family models and the (α,β)-divergence is well-exploited, with explicit updates derived for several important cases (negative binomial, Bernoulli, and Jensen-Shannon).

3- Experiments are thorough: three diverse real-world datasets, multiple loss functions (varying α, β), comparisons against both alternative algorithms (Adam with 6 learning rates) and alternative models (CP, Tucker variants, tensor-train), with careful control for parameter count.

4- Statistical significance is addressed via standard errors, and multiple random initializations are used.

Weaknesses:

1- Not all loss functions satisfy the decomposability property.

2- The reliance on domain knowledge for custom model design.

---

> ### Author Rebuttal · Authors · 2026-03-31
>
> Thank you for your considerate review and strong feedback. We are glad that you appreciate the thoroughness of our experiments and how our method provides a unified formulation and algorithm for general NTF. We address your concerns point-by-point below with a mix of discussion and new empirical results, and have revised the manuscript to reflect our changes.
>
> > 1- Not all loss functions satisfy the decomposability...
>
> > 1- The requirement that loss functions decompose...
>
> While not all loss functions satisfy the decomposability property, almost all loss functions traditionally/often used for NMF/NTF do. Moreover, the decomposability property generalizes loss functions beyond αβ-divergences to an even broader family. **We view a major advantage of the decomposability property (12) as providing a straightforward way to extend to L1 regularization for many loss functions including most $β$ divergences, as often including an L1 penalty preserves the decomposability structure in (12).** Viewing the optimization problem as MLE and extending to MAP estimation corresponds to other forms of regularization (e.g. L2). The framework also lets practitioners quickly derive NTF methods for overdispersed and binary data (Appendix B).
>
> While some divergences violate the decomposability property, we may minimize them indirectly using existing classical inequalities. For example, we may minimize the total variation distance by minimizing the KL divergence directly, as Pinsker's inequality states that $TV(X, Y) \leq \sqrt{\tfrac{1}{2} KL(X,Y)}.$
>
> > 2- The reliance on domain knowledge...
>
> > 3- The custom model design relies on domain expertise...
>
> Currently, practitioners are limited to a small set of models which consists mostly of nonnegative CP/Tucker methods. While it is true that model selection is still an open problem using our framework, it is strictly better to be able to fit a much wider family of models that one may currently do in practice. In most cases, there is at least minimal domain knowledge available. If not, a good starting point is CP/Tucker or even using the observed tensor's dimensions as done in section 6. Our framework directly leverages domain knowledge when possible to rapidly adjust the tensor decomposition model through the modification a single einsum string. This interface allows for the user to iterate quickly and leverage domain knowledge when possible. That being said, **common automatic structure learning approaches can be readily combined with our framework through the tight connection between NTF and discrete density estimation**, which has been recently explored in the literature (Ghalamkari et al. 2025). A future direction of our work is to explore this intersection. For example, model scoring (Cemgil et al. 2019) or cross-validation approaches may be used to determine the most appropriate factorization structure for the given dataset. Our synthetic experiments (see our response to Reviewer joBr) demonstrate the potential efficacy of a cross-validation-based approach. We have revised the paper to further elaborate on this topic in section 7.
>
> > 1- Theoretical results guarantee... (convergence rate)
>
> While we do not directly study the convergence rate in this paper, we suspect that the convergence rate is problem-dependent. In some cases, such as squared Euclidean loss, our updates can be viewed as an adaptive gradient-based method. Here, the convergence rate matches gradient descent. More generally, the convergence rate will depend on the Lipschitz constant and convexity of the loss function.
>
> > 2- In practice, how should practitioners choose α and β...
>
> This is a great question. The user may tune $α$ via cross-validation, as it is a robustness parameter. Prior work (Yilmaz and Cemgil 2012, Simsekli et al. 2013) have studied estimating beta in the setting of beta divergence ($α = 1$). More generally, cross-validation approaches may be an effective tool for selecting both alpha and beta. Specifically, if a user has a specific evaluation criteria in mind (e.g., RMSE, MAE, AUC), then standard cross-validation approaches directly apply.
>
> > 2- The multiplicative updates require computing gradients...
>
> Thank you for highlighting the scalability of our method in practice. **Often, the einsum operation reduces the computational complexity and overhead for operations traditionally ubiquitous in NTF methods, such as computing the matricized tensor times Khatri-Rao product (MTTKRP).** Moreover, we highlight that this is a first-order method that does not require computing higher-order, computationally prohibitive terms like the Hessian, and that direct gradient descent via automatic differentiation-based methods, such as Adam, also involves computing the einsum to evaluate the loss.
>
> Thank you again for your time and consideration. We hope that we have adequately addressed your concerns, which have substantially improved the manuscript. If not, please let us know how we may improve.

---

> > ### Author Rebuttal · Reviewer_zE2D · 2026-04-01
> >
> > Thanks to the authors for their detailed responses. They have addressed my main questions and concerns, including:
> >
> > 1- Clarification on the decomposability property
> >
> > 2- The role of domain knowledge and model selection
> >
> > 3- Theoretical guarantees on convergence rate
> >
> > 4- Guidance on choosing α and β
> >
> > 5- Scalability and computational efficiency
> >
> > I appreciate the thoroughness of their responses.

---

> > > ### Author Response · Authors · 2026-04-02
> > >
> > > Thank you for your quick reply! If you feel we have adequately addressed all your concerns, we would really appreciate it if you would update your scores to properly reflect that. Thanks again for all your feedback and consideration.

---

### Official Review · Reviewer_qRmh · 2026-03-12

**Soundness:** 3
**Presentation:** 3
**Significance:** 2
**Originality:** 3
**Overall Recommendation:** 4
**Confidence:** 2

**Summary:**

This paper proposed an einsum-based multiplicative update algorithm: NNEinFact. The proposed method can easily design a specific tensor decomposition model by changing two parameters.  They also give the theoretical guarantees for the NNEinFact algorithm. Experiments on several real-world tensor datasets demonstrate the effectiveness of the proposed method compared with gradient-based optimization using Adam.

**Compliance With Llm Reviewing Policy:**

Affirmed.

**Final Justification:**

The paper proposes a general and flexible framework for nonnegative tensor factorization with solid theoretical support and practical relevance. My main concerns (modern baselines, optimization comparisons, and structure learning discussion) have been adequately addressed in the rebuttal. I have therefore increased my evaluation and lean towards acceptance.

**Key Questions For Authors:**

see weaknesses.

**Limitations:**

yes

**Strengths And Weaknesses:**

# strengths:
1. The writing is clear, and readers can easily follow.
2. This design could be useful for practitioners who want to explore customized tensor factorization models.
3. The proposed method convergence has the theoretical guarantees.

# weaknesses:
1. The proposed framework mainly demonstrates several classical tensor structures, including CP, Tucker, and Tensor Train. However, recent tensor decomposition models such as Tensor Ring (TR), Fully Connected Tensor Network (FCTN), and Tensor Wheel (TW) have shown improved expressive power in many applications. The experiments do not include comparisons with these more advanced structures, which makes it difficult to assess whether the proposed framework can provide competitive performance against modern tensor network models.
2. The paper suggests that users can design customized tensor structures using the proposed einsum formulation. However, the effectiveness of a manually designed structure may strongly depend on the specific dataset. There has already been a growing body of work on automatic tensor structure search or adaptive model selection. It would be helpful if the paper discussed how the proposed framework relates to these works or whether it can be extended to support automatic structure discovery.
3. The experimental comparison mainly evaluates the proposed multiplicative update (MU) algorithm against Adam with automatic differentiation. However, in the tensor decomposition literature, there exist several specialized optimization methods, such as Alternating Least Squares (ALS) or block coordinate descent (BCD), which are widely used due to their efficiency and strong empirical performance. Since the proposed MU method is also an alternate updating the parameters model, it would be more convincing to compare it with these established optimization algorithms. In particular, for high-order tensor decomposition problems, alternating optimization methods often scale better than gradient-based approaches that update all parameters simultaneously.

---

> ### Author Rebuttal · Authors · 2026-03-31
>
> Thank you for your careful review and outstanding feedback. We are glad you found the manuscript easy to follow and that you recognize the potential usefulness of our method. At a high level, your feedback has greatly improved the manuscript by directing us towards more modern methods, imploring us to show strong improvement over a wider set of baseline methods, and discuss automatic structure recovery. We address each of your concerns point-by-point below.
>
> > The proposed framework mainly demonstrates...
>
> Thank you very much for bringing more modern tensor decomposition models to our attention. **These modern approaches (TR, TW, and FCTN) are actually all specific instances of NNEinfact. As such, an auxiliary contribution of this paper, to the best of our knowledge, is the first nonnegative-constrained implementations for these decompositions.** We have revised the paper to discuss and reference these methods in section 2 (where we also describe how CP, Tucker, tensor-train, and other factorizations are specific instances of NNEinFact). **We have also included tensor ring and tensor wheel as baselines, which we outperform, in the synthetic experiments** (see our response to Reviewer joBr), where we randomly generate custom tensor network structures and attempt to reconstruct the observed tensor using a fixed parameter budget.
>
> > The paper suggests that users can design customized tensor structures... (automatic structure discovery)
>
> This is a great point, automatic structure discovery is a very interesting and relevant topic to this paper given our method's flexibility to fit a wide range of tensor decomposition models. While our work does not focus on automatic structure learning, **common automatic structure learning approaches can be readily combined with our framework through the tight connection between NTF and discrete density estimation**, which has been recently explored in the literature (e.g., Cemgil et al. 2019, Ghalamkari et al. 2025). A future direction of our work is to explore this intersection. For example, model scoring (Cemgil et al. 2019) or cross-validation approaches may be used to determine the most appropriate factorization structure for the given dataset. Our synthetic experiments (see our response to Reviewer joBr) demonstrate the potential efficacy of a cross-validation-based approach. We have revised the paper to further elaborate on this topic in section 7, "Discussion and Conclusion".
>
> Currently, practitioners are limited to a small set of models which consists mostly of nonnegative CP and Tucker methods. While it is true that model selection is still an open problem using our framework, it is strictly better to be able to fit a much wider family of models that one may currently do in practice. In almost all cases, there is some, even if minimal, domain knowledge available. If not, a good starting point is Tucker or CP, or even using the observed tensor's dimensions to form a starting point, as done in section 6. Our framework directly leverages domain-specific information, when possible, to rapidly adjust the tensor decomposition model through the modification a single einsum string. This interface allows for the user to iterate quickly and leverage domain-specific information when possible.
>
> > The experimental comparison mainly evaluates the proposed multiplicative update (MU) algorithm against Adam... (ALS & BCD)
>
> We are glad that you recognize the benefits of BCD approaches and we thank you for bringing the HALS baselines to our attention. The previous literature establishes that if the MU updates constitute an MM algorithm, then they also constitute a BCD method; thus our framework provides a BCD method for any nonnegative tensor network under a wide range of divergences. **Regarding HALS, we have added CP-HALS and Tucker-HALS as baselines and report their performance for each of the UBER, ICEWS, and WITS datasets in our response to Reviewer joBr; they are consistently outperformed by NNEinFact's custom models.** To the best of our knowledge, these are the only existing HALS methods for nonnegative tensor decompositions. Interestingly enough, we find that because the MU are able to directly minimize αβ-divergences while HALS cannot, MU often outperforms HALS in this setting. The inclusion of these baselines has significantly bolstered the claims in our paper, so we thank you for the suggestion.
>
> Thank you very much for your time and consideration. Your feedback has greatly improved the quality of the manuscript, and we hope that we have adequately addressed your concerns. If not, please let us know how we may further improve.

---

> > ### Author Rebuttal · Reviewer_qRmh · 2026-04-02
> >
> > The authors have carefully addressed my concerns. In particular:
> >
> > - They clarified the relation to modern tensor network models (TR, TW), and included them as baselines with empirical comparisons.
> > - They expanded the discussion on automatic structure learning and clearly positioned it as a future direction, with concrete connections to existing approaches.
> > - They incorporated additional comparisons with BCD-style methods (e.g., HALS).
> >
> > Overall, these significantly strengthen the paper and improve its positioning within the tensor decomposition literature. I will increase my score.

---

### Official Review · Reviewer_joBr · 2026-03-13

**Soundness:** 3
**Presentation:** 3
**Significance:** 3
**Originality:** 3
**Overall Recommendation:** 4
**Confidence:** 3

**Summary:**

This paper addresses a gap in nonnegative tensor factorization (NTF) by providing a general framework for deriving Multiplicative Update (MU) rules. The authors demonstrate that any NTF model expressible via einsum notation can be optimized under a wide range of $( \alpha, \beta )$-divergence losses, provided they satisfy a convex-concave decomposability condition.
Experiments on Uber (NYC taxi), ICEWS (conflict events), and WITS (trade flows) compare MU against Adam with six learning rates.
NNEinFact converges faster in wall-clock time and achieves lower heldout loss across settings, and custom einsum models outperform standard CP/Tucker baselines. The framework covers CP, Tucker, Tensor Train, and arbitrary custom factorizations within a single algorithm.

**Compliance With Llm Reviewing Policy:**

Affirmed.

**Key Questions For Authors:**

See above

**Limitations:**

Yes

**Strengths And Weaknesses:**

## Strengths

- **Practical generality.** Einsum notation is an elegant abstraction — the algorithms requires only a model string and swap operator, making it easy to define custom factorizations without reimplementing update rules.
- **Empirical efficiency.** Results show consistent convergence advantages over Adam: MU reaches lower heldout loss faster across four of six (α,β) settings on Uber,and ICEWS/WITS.
- **Convergence guarantee.** Authors provide theoretical results showing  NNEinFact converges to a local minimum of the loss landscape.
---

## Major Concerns

- **The only baseline is Adam — specialized MU and ALS methods are excluded.**
The paper compares NNEinFact's MU only against Adam (autodiff). For standard NMF and CP decomposition, hierarchical ALS (HALS) and specialized MU implementations (e.g., Cichocki et al. 2011, which the paper already cites) are well-established alternatives that often outperform gradient descent.
- **No simulation recovery experiment.** All evaluations use real data with unknown ground truth. A synthetic experiment (with known latent factors) would validate that MU finds the correct solution, not just a low-loss point.
---

---

> ### Author Rebuttal · Authors · 2026-03-31
>
> Thank you for your carefully review and helpful feedback. We are glad that you find the einsum notation to be an elegant abstraction for NTF and appreciated our theoretical and empirical results. **Your concerns motivated us to conduct additional real-world data experiments which demonstrate our method's superiority over two ALS baselines and design an exhaustive set of synthetic experiments, which show NNEinFact's superior modeling, ground-truth recovery, and predictive capabilities**. These additions have significantly strengthened the manuscript. We address your concerns point-by-point below.
>
> > The only baseline is Adam — specialized MU and ALS methods are excluded...
>
> Thank you for pointing us to compare NNEinFact to modern methods. We use automatic differentiation as a baseline in Figure 2 since it is the only method which allows for such tailored, custom models to be fit. In the cases of CP and Tucker decomposition, the specialized MU implementation of Cichocki et al. corresponds very closely to our αβ divergence updates. **To address your concern, we have added CP-HALS and Tucker-HALS as a baselines in Figure 3 and Table 3, (where Cichocki's CP and Tucker's multiplicative update schemes are already baselines).** We note an interesting phenomenon: since HALS methods minimize Euclidean distance rather than directly minimizing the αβ-divergence, they tend to perform even worse than the specialized MU-update CP and Tucker schemes (as measured by heldout loss).
>
> Mean heldout loss, identical to table 3 with CP-HALS and Tucker-HALS added  (standard error in parentheses):
>
> | Dataset | $α$ | Custom | CP-HALS | Tucker-HALS |
> |---|---:|---:|---:|---:|
> |  | 0.8 | **0.0080** (0.00002) | 0.0094 (0.00002) | 0.014 (0.0008) |
> | Uber | 1.0 | **0.010** (0.00003) | 0.012 (0.00003) | 0.022 (0.003) |
> |  | 1.2 | **0.015** (0.00007) | 0.018 (0.0002) | 0.040 (0.003) |
> |  | 0.8 | **0.020** (0.00003) | 0.025 (0) | 0.031 (0) |
> | ICEWS | 1.0 | **0.027** (0.0001) | 0.033 (0) | 0.044 (0) |
> |  | 1.2 | **0.046** (0.0001) | 0.058 (0) | 0.074 (0) |
> |  | 0.8 | **0.17** (0.004) | 0.29 (0.005) | 0.50 (0.002) |
> | WITS | 1.0 | **0.036** (0.002) | 0.066 (0.002) | 0.18 (0.001) |
> |  | 1.2 | **0.013** (0.0009) | 0.028 (0.001) | 0.11 (0.001) |
>
>
> > No simulation recovery experiment...
>
> Thank you for suggesting that we run synthetic experiments to validate our method. We have designed a set of synthetic experiments and have revised the paper to include them in the appendix. In particular, we select 3 non-traditional tensor network structures and generate data from them. To compare to CP and Tucker baselines, we consider settings with ground truth factor matrices. Varying the sparsity level, we compute the cosine similarity of the estimated factors to the ground truth, repeating each experimental setting 10 times. The mean and standard errors are reported in the table below.
>
> |% sparse|Custom|CP|Tucker|
> | ---: | ---: | ---: | ---: |
> |0.3|**0.99** (0.005) |0.63 (0.003) |0.93 (0.01) |
> |0.5|**0.96** (0.01) |0.59 (0.01) |0.91 (0.02) |
> |0.7|**0.92** (0.02) |0.51 (0.01) |0.83 (0.02) |
> |0.9|**0.85** (0.02) |0.50 (0.01) |0.65 (0.04) |
>
>  We ran another set of synthetic experiments to evaluate our method's reconstruction capabilities when the ground truth is known and matches the fitted model. For  200 randomly-generated tensor network structures, we show that NNEinfact consistently achieves better recovery, as measured by relative RMSE, than other tensor decomposition structures, including more modern ones (tensor train, tensor ring and tensor wheel), which cannot recover the true underlying structure due to model misspecification. Such model misspecification penalizes the reconstruction ability of these baseline methods, as shown in the table below. We find that empirically, the recovery discrepancy increases with the number of modes.
>
> Synthetic results: Mean reduction in relative RMSE (baseline - NNEinFact relative RMSE: **greater than zero implies improvement over baseline**):
>
> | Modes: | 3 | 4 | 5 | 6 | 7 | 8 |
> | --- | ---: | ---: | ---: | ---: | ---: | ---: |
> | CP |−0.02 (0.01) |0.05 (0.03) | 0.03 (0.03) | 0.10 (0.03) | 0.09 (0.05) | 0.20 (0.05) |
> | Tucker |0.04 (0.04) |0.15 (0.04) | 0.18 (0.06) | 0.34 (0.08) | 0.68 (0.20) | 0.89 (0.15) |
> | Tensor Train |0.03 (0.06) |0.19 (0.05) | 0.15 (0.06) | 0.37 (0.08) | 0.52 (0.09) | 0.91 (0.14) |
> | Tensor Ring |0.02 (0.03) |0.14 (0.04) | 0.19 (0.06) | 0.30 (0.08) | 0.48 (0.07) | 0.94 (0.13) |
> | Tensor Wheel |−0.01 (0.02) |0.04 (0.02) | 0.08 (0.03) | 0.11 (0.03) | 0.35 (0.17) | 0.53 (0.12) |
>
> We provide example model strings below:
>
> - 4 modes: `aP,bP,cQS,dRS,PQRS->abcd`
> - 6 modes: `aPQR,bPQS,cPRT,dQRT,ePST,fQST,PQRST->abcdef`
> - 8 modes: `aPQR,bPQS,cPRT,dQRT,ePSU,fQTU,gRSV,hTUV,PQRSTUV->abcdefgh`
>
> We thank you for your time and insightful comments, which have greatly improved the manuscript. We hope that our response adequately addresses your concerns, and if not, please let us know how we may improve.

---

> > ### Author Rebuttal · Reviewer_joBr · 2026-04-02
> >
> > Thanks for the response. Authors have addressed my concerns about baseline and simulation studies.
> > I will maintain my initial score

---

> > > ### Author Response · Authors · 2026-04-02
> > >
> > > Thanks for your quick reply. We believe the work we have done to address your feedback, detailed above, has greatly strengthened the paper. If you agree that this has addressed your initial concerns, then we ask that you update your score to properly reflect that fact. Thanks again for your very useful feedback.

---

### Official Review · Reviewer_vswQ · 2026-03-15

**Soundness:** 3
**Presentation:** 3
**Significance:** 2
**Originality:** 2
**Overall Recommendation:** 4
**Confidence:** 4

**Summary:**

This paper introduces NNEinFact for nonnegative tensor network decomposition.  NNEinFact employs an MM framework to derive robust multiplicative updates. A core theoretical contribution is the establishment of a convex-concave decomposability condition, which guarantees that the algorithm monotonically converges to a local minimum across a diverse family of user-specified loss functions, notably the $(\alpha, \beta)$-divergences. Experimental results show the convergence advantages of the proposed method compared to Adam.

**Compliance With Llm Reviewing Policy:**

Affirmed.

**Final Justification:**

I appreciate the detailed rebuttal of the authors in the previous two-round discussions. The authors have reclaimed and re-emphasized the contributions of this paper. These days, I have re-read this paper and tried to figure out all the details, so as to provide a convincing justification after the rebuttal.

From the contribution side, I really appreciate that the authors derive these near-universal MU rules for fo the einsum decomposition, which may broadly contribute to the nonnegative matrix/tensor decomposition. The experimental evaluation and empirical evidence also demonstrated the advantages.

However, as the general ($\alpha, \beta$) divergence for NMF was developed many years ago, it is not a very surprising result to generalize to a generic nonnegative tensor decomposition using multiplicative update rules. On the MM-side, I think the authors developed it with thought by designing the $Q(\Theta | \tilde{\Theta})$ as it is quite appropriate for such a universal nonnegative optimization problem.

Overall, I think this paper is somewhat sound, and I would like to increase my rating to weak acceptance.

**Key Questions For Authors:**

1. Regarding the theoretical guarantees, it would be better to provide more discussions on the technical novelty of the results or proof details, since the MM framework is quite common.
2. Can the authors provide more experiments on different TN structures such as some random generated TN structures to show the generality of the proposed algorithm?
3. The gradient-based optimization is very sensitive to the initialization especially for gradient-based tensor decomposition. Therefore, it would be better to investigate different intialization strategies for Adam-based algorithm, which would be more fair.
4. In this optimization framework, it would be better to add the extension discussion on how to incorporate some regularization terms such as sparsity or orthogonal terms in this optimization framework, since vanilla nonnegative TN may not be good enough for some applications.

**Limitations:**

See weakness above.

**Strengths And Weaknesses:**

**Strengths**
1. This paper is clearly and well written, and the technical contribution is clear.
2. This paper introduces a nearly universial MU-based algorithm for optimizing nonnegative tensor networks (TNs). The framework is general and can be applied to a wide range of TN structures and loss functions.


**Weaknesses**

1. The optimization for ($\alpha$, $\beta$) is quite common, whihch is not a significant contribution since it has been widely used in the literature and proposed in many years ago. The extension to a general TN structure is not a significant contribution, since it is a straightforward extension of the existing MU-based algorithms for NMF.
2. The authors introduced an MM-based optimization algorithm, and proved its convergence. However, the proof framework is also quite common, which is not a significant contribution.

---

> ### Author Rebuttal · Authors · 2026-03-31
>
> Thank you for carefully reviewing our work and providing useful feedback. We are glad you find the paper clear and well-written and that you appreciated the significance of some of its theoretical contributions, such as the decomposability condition.
>
> We want to emphasize that **our core contributions are methodological**. While it is true that αβ-divergences have been presented, analyzed, and optimized in prior work, there is nevertheless still a large gap in the literature in terms of general and reliable methodological frameworks that allow practitioners to flexibly fit a wide range of models under any such divergences. We agree that the theoretical components of our paper build directly upon existing work, and do not themselves represent a major leap. However, those theoretical advances are intended to support the methodological advances that constitute the paper's main contribution. In response to your suggestion, we have revised the paper to make these points clearer and to provide more context for the paper's theory.
>
> Moreover, we want to highlight that **the decomposability condition allows us the framework to extend well beyond αβ-divergences, and in response to your suggestions, we have used this fact to extend the framework to incorporate MAP estimation under conjugate priors (including L2-reg. under the squared Euclidean loss) as well as other forms of regularization (e.g. L1)**.
>
> In response to your substantive feedback on the methodological framework, we have also designed many new suites of experiments that demonstrate many things you have asked about, most notably:
> * the proposed framework recovers ground-truth parameters across a wide range of random tensor structures, and
> * the comparison to baselines is robust to different initialization schemes.
>
> We thank you for these suggestions; implementing them has greatly improved the paper. Please let us know of any further improvements.  In what follows, we address your comments one-by-one below.
>
> > The optimization for αβ-divergences is quite common...
>
> While minimizing αβ-divergences for NMF/CP/Tucker is known, the decomposability property (Eq. 12) is novel (to the best of our knowledge). It is a general property that readily derives methods for non-traditional loss functions (Appendix B). However, our primary contribution is algorithmic: the unprecedented ability to specify and fit an arbitrary tensor decomposition with a single string. We view it as a strength that this flexible tool generalizes existing work while maintaining convergence guarantees.
>
> > The authors introduced an MM-based optimization algorithm...
>
> While MM is common, it has not been paired with the einsum operation. Their coupling is a substantial practical and algorithmic contribution. It is great that existing proof techniques bring rigor to our method without compromising flexibility.
>
> > Regarding the theoretical guarantees...
>
> We have revised the paper to include additional discussion in section 2 highlighting the novelty and utility of the decomposability condition (eq. 12), which has not appeared in the literature, to the best of our knowledge.
>
> > Can the authors provide more experiments...
>
> We have run many synthetic experiments to address this; please refer to our response to Reviewer joBr.
>
> > The gradient-based optimization is very sensitive to the initialization...
>
> We tested two additional initialization strategies (standard log-normal and standard exponential) using the best initial learning rates (0.1, 0.3, 0.5) from the original scheme. Reporting the best-performing combination's heldout loss below, the table shows **our method outperforms the baseline across initializations**.
>
> |Dataset|α,β|Einfact|Adam|
> |---|---:|---:|---:|
> |UBER|0.7,0.0|**0.008**|0.010|
> ||0.7,1.0|**0.043**|0.190|
> ||1.0,0.0|**0.011**|0.015|
> ||1.0,1.0|**0.342**|0.46|
> ||1.3,0.0|**0.022**|0.029|
> ||1.3,1.0|**0.862**|0.896|
> |ICEWS|0.7,0.0|**0.018**|0.028|
> ||0.7,1.0|**0.093**|0.152|
> ||1.0,0.0|**0.031**|0.050|
> ||1.0,1.0|**0.756**|1.167|
> ||1.3,0.0|**0.070**|0.114|
> ||1.3,1.0|**2.035**|2.989|
> |TRADE|0.7,0.0|**0.009**|**0.009**|
> ||0.7,1.0|0.095|**0.094**|
> ||1.0,0.0|**0.013**|**0.013**|
> ||1.0,1.0|**0.803**|3.623|
> ||1.3,0.0|**0.031**|**0.031**|
> ||1.3,1.0|**2.172**|11.117|
>
> While Adam is gradient-based, we chose it because of its strong empirical performance and robustness to initial conditions, due to its adaptive learning rate and bias correction. **We also added HALS-CP/Tucker as baselines, which our method outperforms** (see response to Reviewer joBr).
>
> > In this optimization framework...
>
> Extensions to sparsity are particularly applicable here. **A strength of the decomposability property (eq. 12) is that it straightforwardly induces sparsity**: adding an L1 term often preserves (12). Similarly, (12) is preserved in cases of conjugate priors in MAP estimation, enabling other regularization (e.g. L2, Euclidean loss). The nonnegative nature of the domain prohibits an adaptation to orthogonality.

---

> > ### Author Rebuttal · Reviewer_vswQ · 2026-04-03
> >
> > I appreciate the authors' response. In this response, the authors clarified the contributions of this paper and provided additional experiments.
> >
> > However, I still think this paper is overall technically limited, as many components are very well-known techniques. The main contributions of this paper are to extend it to a tensor decomposition case, or more generally, to an einsum factorization. While this implementation is somewhat meaningful, I am not yet fully convinced that it constitutes a sufficiently technical advance, since matrix factorization is a basic instance of einsum contraction. From this perspective, the proposed $(\alpha,\beta)$-divergence and the MM-based optimization appear to be relatively incremental.
> >
> > In this case, I would like to maintain my original rating.

---

> > > ### Author Response · Authors · 2026-04-06
> > >
> > > Thank you for your critical feedback. We believe that extending the framework from matrix factorization to any multilinear tensor contraction is actually a very substantial contribution which vastly widens the family of models that practitioners can now easily build and fit. Moreover, we want to re-emphasize that, in addition to extending the framework to all multilinear forms under the $(\alpha, \beta)$-divergence, deriving all results under the decomposability property actually enables optimization beyond the $(\alpha, \beta)$-divergence. Thank you again for your consideration.

---

### Decision · Program_Chairs · 2026-04-30

**Decision:**

Accept (regular)

**Comment:**

After considering the reviews, rebuttal, and the reviewer discussion, I recommend acceptance. The discussion was important in reaching a clearer consensus: concerns about novelty remain and are legitimate (reliance on well-established ingredients), the exchange among reviewers helped clarify that the contribution is still meaningful and potentially quite valuable to the community. In particular, the paper provides a practical and comprehensive framework for nonnegative einsum factorization that unifies a broad class of models in a way that should be useful to both researchers and practitioners.

In the end, the paper offers a useful contribution that the community is likely to build on, hence the weak accept recommendation.